# LPP is a Src substrate required for invadopodia formation and efficient breast cancer lung metastasis

Elaine Ngan[1,2], Konstantin Stoletov[3], Harvey W. Smith[1,4], Jessica Common[1,2], William J. Muller[1,2,4], John D. Lewis[3] & Peter M. Siegel[1,2,4,5]

We have previously shown that lipoma preferred partner (LPP) mediates TGFβ-induced breast cancer cell migration and invasion. Herein, we demonstrate that diminished LPP expression reduces circulating tumour cell numbers, impairs cancer cell extravasation and diminishes lung metastasis. LPP localizes to invadopodia, along with Tks5/actin, at sites of matrix degradation and at the tips of extravasating breast cancer cells as revealed by intravital imaging of the chick chorioallantoic membrane (CAM). Invadopodia formation, breast cancer cell extravasation and metastasis require an intact LPP LIM domain and the ability of LPP to interact with α-actinin. Finally, we show that Src-mediated LPP phosphorylation at specific tyrosine residues (Y245/301/302) is critical for invadopodia formation, breast cancer cell invasion and metastasis. Together, these data define a previously unknown function for LPP in the formation of invadopodia and reveal a requirement for LPP in mediating the metastatic ability of breast cancer cells.

[1] Goodman Cancer Research Centre, McGill University, Montreal, Quebec, Canada H3A 1A3. [2] Department of Medicine, McGill University, Montreal, Quebec, Canada H4A 3J1. [3] Department of Oncology, University of Alberta, Edmonton, Alberta, Canada T6G 2E1. [4] Department of Biochemistry, McGill University, Montreal, Quebec, Canada H3G 1Y6. [5] Department of Anatomy and Cell Biology, McGill University, Montreal, Quebec, Canada H3A 0C7. Correspondence and requests for materials should be addressed to P.M.S. (email: peter.siegel@mcgill.ca).

nvadopodia are critical structures employed by cancer cells to intravasate into the bloodstream and extravasate into secondary sites during the metastatic process[1]. They are located on the ventral side of invading cancer cells and are rich in actin-containing complexes that include: WASP, Arp2/3, Cortactin, Tks4/5 and c-Src (refs 2–7). Furthermore, they possess the ability to locally degrade extracellular matrix (ECM) via the activity of diverse proteases including: MMP2, MMP9, MT1-MMP, ADAM12, ADAM15, and ADAM19 (ref. 8). Invadopodia allow cancer cells to escape the primary tumour, breach vascular barriers and colonize distant organs[9,10]. Recent advances in live cell imaging permit the visualization of these structures *in vivo* during intravasation and extravasation[11–13] and reveal that cancer cells engage invadopodia to breach the endothelium during the earliest stages of the metastatic process. Moreover, inhibition of these structures significantly diminishes tumour cell extravasation and the formation of breast cancer metastases[13,14]. In this regard, TGFβ promotes Src-induced invadopodia formation via Hic-5 upregulation, while knockdown of Twist1, a central mediator of EMT, abrogates their formation[15,16]. Collectively, these data emphasize a role for a TGFβ-induced EMT in promoting invadopodia formation and metastasis.

We have previously characterized lipoma preferred partner (LPP) as a critical mediator of TGFβ-induced cell migration and invasion in breast cancer cells capable of undergoing an EMT[17]. LPP is a member of the zyxin family of proteins that regulates cytoskeletal organization, cell motility and mechanosensing[18,19]. Following TGFβ stimulation, we demonstrated that LPP localizes to focal adhesions via its LIM1 domain and recruits α-actinin to stress fibres as a mechanism to promote migration and invasion of mammary tumour cells[17]. In this context, LPP enhances focal adhesion dynamics within ErbB2-expressing breast cancer cells[17]. In the current study, we delineate an important role for LPP as a Src substrate, a positive regulator of invadopodia formation and an enhancer of breast cancer metastasis.

## Results

**LPP is a critical mediator of breast cancer metastasis.** ErbB2 expressing NMuMG cells (NMuMG-ErbB2) spontaneously metastasize to the lung from the primary tumour and efficiently form lung metastases following tail vein injection[20,21]. Using this system, we previously demonstrated that LPP promotes the *in vitro* migration and invasion of breast cancer cells following a TGFβ-induced EMT[17]. To assess the requirement of LPP for breast cancer metastasis *in vivo*, we stably reduced LPP expression levels in NMuMG-ErbB2 cells and the NIC breast cancer model, which was derived from an ErbB2-expressing mammary tumour that arose in transgenic mice expressing an activated form of ErbB2 (ref. 22; Supplementary Fig. 1). Stable loss of LPP does not impair a TGFβ-induced EMT, as demonstrated by the induction of mesenchymal markers (Snail, vimentin, fibronectin and α-SMA) and the loss of the epithelial marker, E-cadherin (Supplementary Fig. 2a,b). Furthermore, reduced LPP levels do not influence breast cancer cell proliferation (Supplementary Fig. 2c,d) or signalling responses induced by TGFβ, including phosphorylation of Smad2 (Supplementary Fig. 2e,f).

We injected NMuMG-ErbB2 and NIC cells expressing a stable shRNA against LPP (LPP-shRNA) into the mammary fat pads of female immunocompromised mice, along with breast cancer cells harbouring a non-targeting shRNA control (LucA-shRNA). Tumour growth was monitored by weekly caliper measurement for 4 weeks, at which time mammary tumours were resected. We did not observe any differences in primary tumour growth in

mice injected with NMuMG-ErbB2 or NIC breast cancer cells harbouring LPP- or LucA-shRNAs (Fig. 1a,b).

To determine if reduced LPP expression affected metastasis, we monitored these cohorts following primary tumour resection. Lungs were harvested 3 weeks post resection for cohorts bearing NMuMG-ErbB2 tumours, and 4 weeks post resection for mice bearing NIC tumours (LucA- and LPP-shRNAs). A clear diminishment in both the number of lung surface lesions and the area of lung tissue occupied by breast cancer metastases was observed in animals injected with breast cancer cells harbouring an LPP-shRNA in both the NMuMG-ErbB2 and NIC models when compared with mice injected with cells containing LucA-shRNA (Fig. 1c,d). IHC staining confirmed that lung metastases derived from LPP-shRNA expressing mammary tumours maintained ErbB2 levels, while exhibiting diminished LPP expression (Supplementary Fig. 3). These data demonstrate that LPP is dispensable for the growth of primary breast tumours but is critical for lung metastasis.

**LPP promotes breast cancer cell intravasation.** Breast cancer cells must intravasate into the circulation to establish distal metastases. To investigate whether LPP promotes breast tumour cell intravasation *in vivo*, we injected NMuMG-ErbB2 and NIC breast cancer cells that possessed LucA- and LPP-shRNAs, into the mammary fat pads of immunocompromised mice. Consistent with our previous observations, loss of LPP was dispensable for mammary tumour growth (Supplementary Fig. 4a–d); however, mice bearing tumours harbouring an LPP-shRNA possessed significantly fewer lung metastases when compared with mice bearing a LucA-shRNA-expressing tumours (Supplementary Fig. 4e,f). We next collected whole blood from these animals, isolated viable circulating tumour cells (CTCs) and quantified the number of CTC-derived adherent colonies after 2 weeks of culture in selection media. Mice bearing NMuMG-ErbB2 tumours harbouring a LucA-shRNA had significantly more CTCs than mice with LPP-shRNA expressing NMuMG-ErbB2 tumours (average of 15.6 ± 3.98 versus 3.59 ± 1.59 colonies, respectively; Fig. 2a). Blood samples collected from animals with LPP-shRNA expressing NIC tumours did not produce any CTC-derived colonies while NIC-tumour bearing mice from the control cohort (Luc-shRNA) produced an average of 123.6 ± 7.8 colonies (Fig. 2b). These data suggest that LPP plays an important role in tumour cell intravasation from the primary tumour into the bloodstream.

**LPP promotes invadopodia formation and ECM degradation.** Invadopodia are specialized degradative structures that facilitate tumour cell intravasation and efficient metastasis[10,13,23]. Given the importance of LPP in mediating these processes, we investigated whether LPP also plays a role in invadopodia formation and ECM degradation. We employed HCC1954 human breast cancer cells, which engage LPP to promote cell migration and invasion in response to TGFβ (ref. 17). Co-localization of Tks5 and F-actin has previously been demonstrated to be a strong indicator of active invadopodia[8,24]. HCC1954 cells, treated with TGFβ and transfected with either scrambled or LPP-targeting siRNAs, were plated onto fluorescently labelled-gelatin for 24 h. Fixed coverslips were stained with antibodies against Tks5 and phalloidin (F-actin) and the area of gelatin degradation (as indicated by the loss of fluorescent signal) was quantified (Fig. 3a and Supplementary Fig. 5a). Our data reveal that siRNA-mediated knockdown of LPP significantly reduced the area of gelatin proteolysis by HCC1954 breast cancer cells (Fig. 3a).

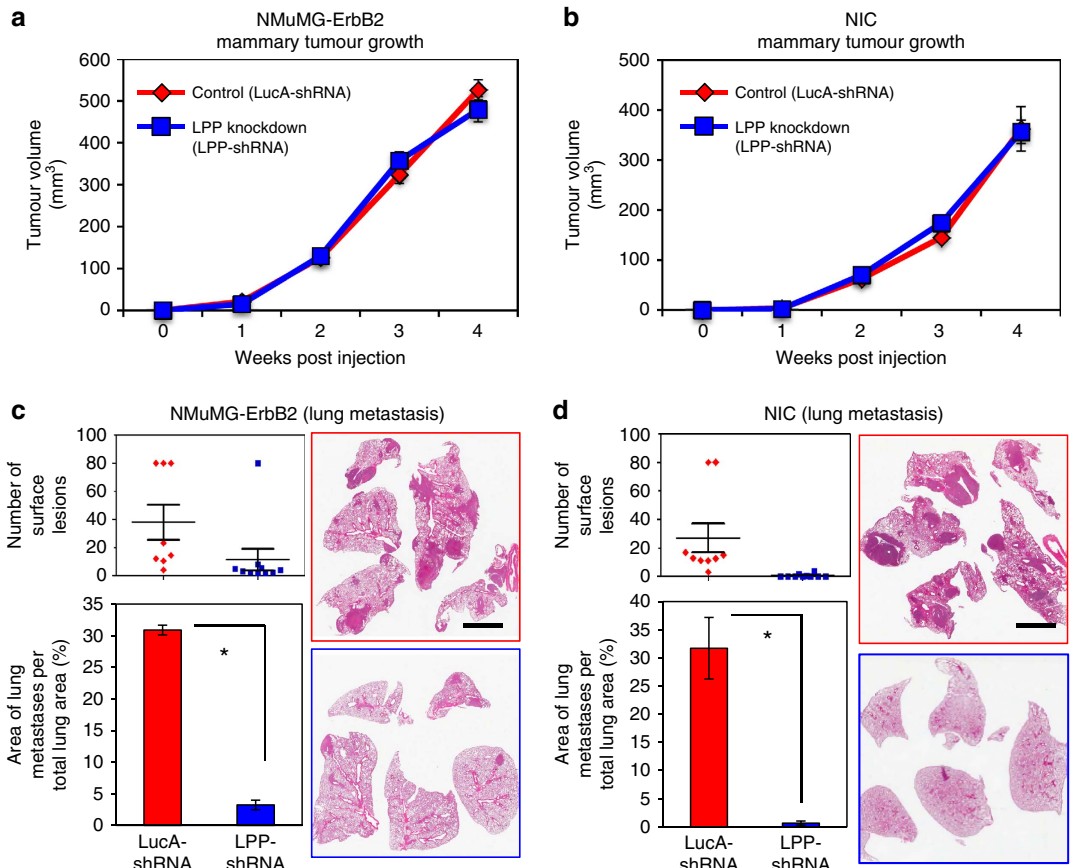

**Figure 1 | LPP is dispensable for mammary tumour growth but is required for the efficient formation of lung metastases. (a,b)** NMuMG-ErbB2 and NIC breast cancer cells harbouring shRNAs targeting LPP (LPP-shRNA) or LucA control (LucA-shRNA) were injected into the mammary fat pads of mice ($n = 10$ per cohort). Primary tumour growth was monitored by weekly caliper measurement and tumours were resected after 28 days. Mice were euthanized 3 (NMuMG-ErbB2) or 4 weeks (NIC) post primary mammary tumour resection and lung tissue was collected at necropsy. **(c,d)** The number of macroscopic lesions on the lung surfaces were quantified at necropsy from cohorts of mice injected with NMuMG-ErbB2 LucA-shRNA ($n = 8$) and NMuMG-ErbB2 LPP-shRNA, ($n = 10$) **(c)** or NIC LucA-shRNA ($n = 9$) and NIC LPP-shRNA ($n = 10$) breast cancer cells **(d)**. The area of metastatic burden was quantified from H&E stained lung sections from the same samples, and is expressed as a percentage of total lung surface area **(c,d)**. *$P < 0.006$. Representative images are shown **(c,d)**. Scale bar, 3 mm and applies to all images in **c** and **d**. Error bars represent s.e.m. for all panels.

We extended our analysis to an ErbB2-expressing mouse breast cancer cell model. TGFβ-treated NMuMG-ErbB2 cells harbouring a LucA-shRNA readily degraded the gelatin substrate, while loss of LPP (LPP-shRNA) abrogated the degradative ability of these cells (Fig. 3b and see Supplementary Fig. 5b). LPP is composed of three carboxy-terminal LIM (LIN-11, ISL-1, MEC-3) domains and a proline-rich amino-terminal region (PRR), within which resides several protein–protein interaction motifs[19]. We have previously demonstrated that the LIM1 domain and the α-actinin-binding domain (ABD) of LPP are important for TGFβ-mediated breast cancer migration and invasion[17]. While expression of eGFP-LPP-WT rescued the degradative phenotype of NMuMG-ErbB2 cells, neither a LIM domain mutant of LPP (eGFP-LPP-mLIM1) nor one that cannot bind to α-actinin (eGFP-LPP-ΔABD) were able to form invadopodia and induce gelatin degradation (Fig. 3b and see Supplementary Fig. 5b). Thus, the ability of NMuMG-ErbB2 breast cancer cells to form invadopodia and to degrade ECM is dependent on LPP expression, and moreover, relies on the presence of an intact LIM1 domain and the ABD. Together, loss of LPP expression or function reduces the ability of multiple breast cancer cell lines to degrade gelatin ECM.

**LPP localizes with Tks5/actin in active invadopodia.** In light of these observations, we next investigated whether LPP was localized

to active invadopodia. HCC1954 and NMuMG-ErbB2 breast cancer cells were stimulated with TGFβ for 24 h, plated onto Alexa 405 labelled gelatin for an additional 24 h and coverslips were fixed and stained for LPP, Tks5 and F-actin. We observed that a subpopulation of LPP, which is also a constituent of focal adhesions[17], was co-localized with Tks5 and F-actin at sites of gelatin degradation in both HCC1954 (Fig. 4a) and NMuMG-ErbB2 cells (Supplementary Fig. 6a). Furthermore, line scan analysis shows the simultaneous presence of LPP with Tks5 and F-actin within sites of ECM proteolysis in both cell models (Fig. 4b and Supplementary Fig. 6b). Z-Stack acquisitions coupled with orthogonal views ($x$–$z$, $y$–$z$) through sites of degradation revealed the co-localization of LPP, Tks5 and F-actin throughout the vertical plane of the gelatin, suggesting their presence within active, matrix-degrading invadopodial protrusions (Fig. 4c and Supplementary Fig. 6c). Collectively, our data clearly show that LPP localizes to invadopodia structures in breast cancer cells that actively degrade matrix.

**Reduced LPP does not impair TGFβ-induced MMP activity.** Matrix metalloproteinases (MMPs), including MMP2, MMP9 and MTI-MMP (MMP14), are critical mediators within invadopodia that promote cancer cell invasion[8]. As previously shown, reducing LPP levels or impairing LPP interactions with the actin cytoskeleton impaired gelatin degradation (Fig. 3); thus,

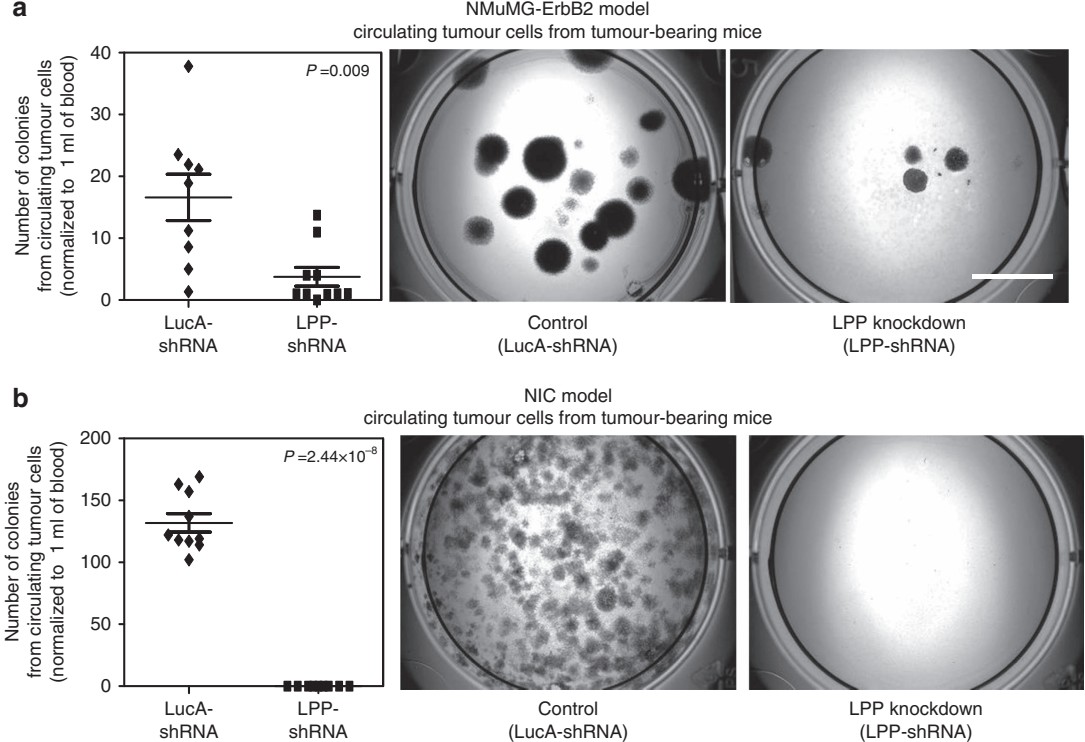

**Figure 2 | Loss of LPP diminishes breast cancer-derived circulating tumour cells.** Whole blood was collected by cardiac puncture from mice-bearing NMuMG-ErbB2 or NIC mammary tumours, harbouring LucA- and LPP- shRNAs, to isolate CTCs. The number of CTC-derived adherent colonies was determined 2 weeks post isolation and were quantified from (**a**) NMuMG-ErbB2 LucA-shRNA ($n = 9$) and NMuMG-ErbB2 LPP-shRNA, ($n = 10$) or (**b**) NIC LucA-shRNA ($n = 10$) and NIC LPP-shRNA ($n = 10$) cultures. Representative images of formalin-fixed and crystal violet stained samples are shown. Scale bar, 1 cm, and applies to all images. Error bars represent s.e.m.

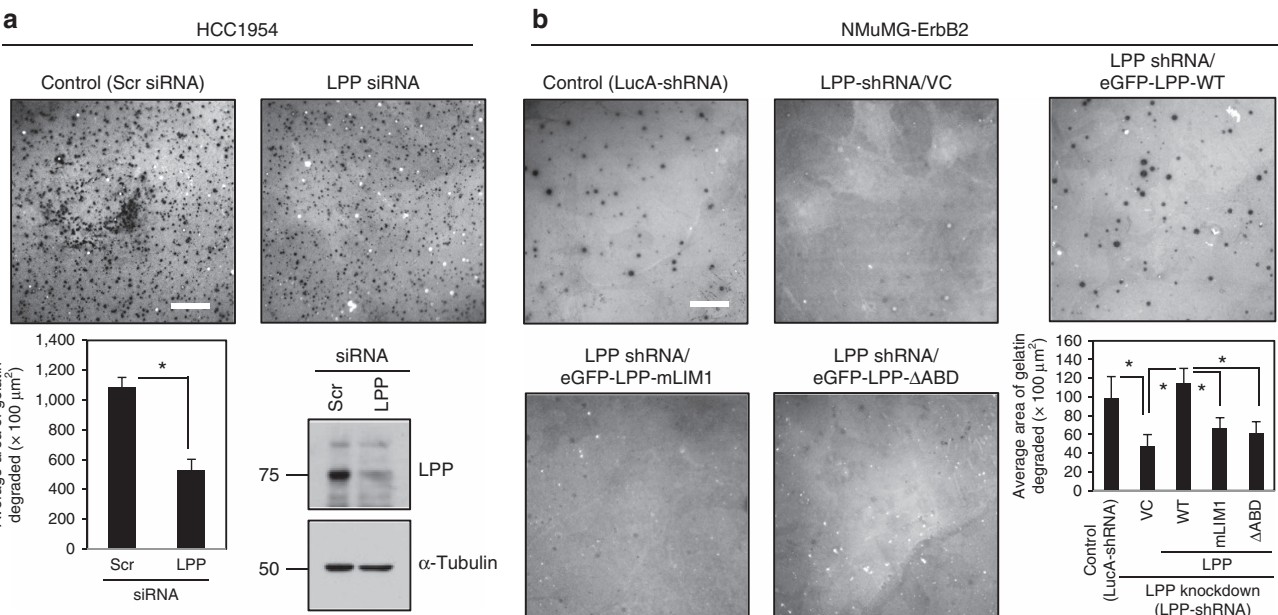

**Figure 3 | LPP promotes breast cancer cell-mediated gelatin degradation.** (**a**) HCC1954 human breast cancer cells were transfected with either scrambled (control) or LPP targeting siRNAs. Cells were pre-treated with TGFβ for 24 h and then plated onto fluorescently labelled gelatin for an additional 24 h. The area of degradation was determined from images taken from three independent experiments (control $n = 34$, LPP siRNA $n = 29$), and error bars represent s.e.m. Immunoblot analysis was performed to assess the level of LPP knockdown. α-Tubulin was used as a loading control. Scale bar, 20 μm, and applies to both images. *$P = 1.11 \times 10^{-6}$. (**b**) NMuMG-ErbB2 cells harbouring shRNA against LucA (LucA-shRNA) or targeting LPP (LPP-shRNA) were used to express an EV (VC) or LPP rescue constructs (eGFP-LPP-WT, eGFP-LPP-mLIM1 and eGFP-LPP-ΔABD). The indicated breast cancer cells were pre-treated with TGFβ for 24 h, and subsequently plated onto fluorescently labelled gelatin for an additional 24 h. The area of degradation was quantified from images of LucA control ($n = 26$), VC ($n = 25$), WT ($n = 27$), mLIM1 ($n = 26$) and ΔABD ($n = 24$) cells from four independent experiments and error bars represent s.e.m. *$P < 0.05$. Scale bar, 20 μm, and applies to all images.

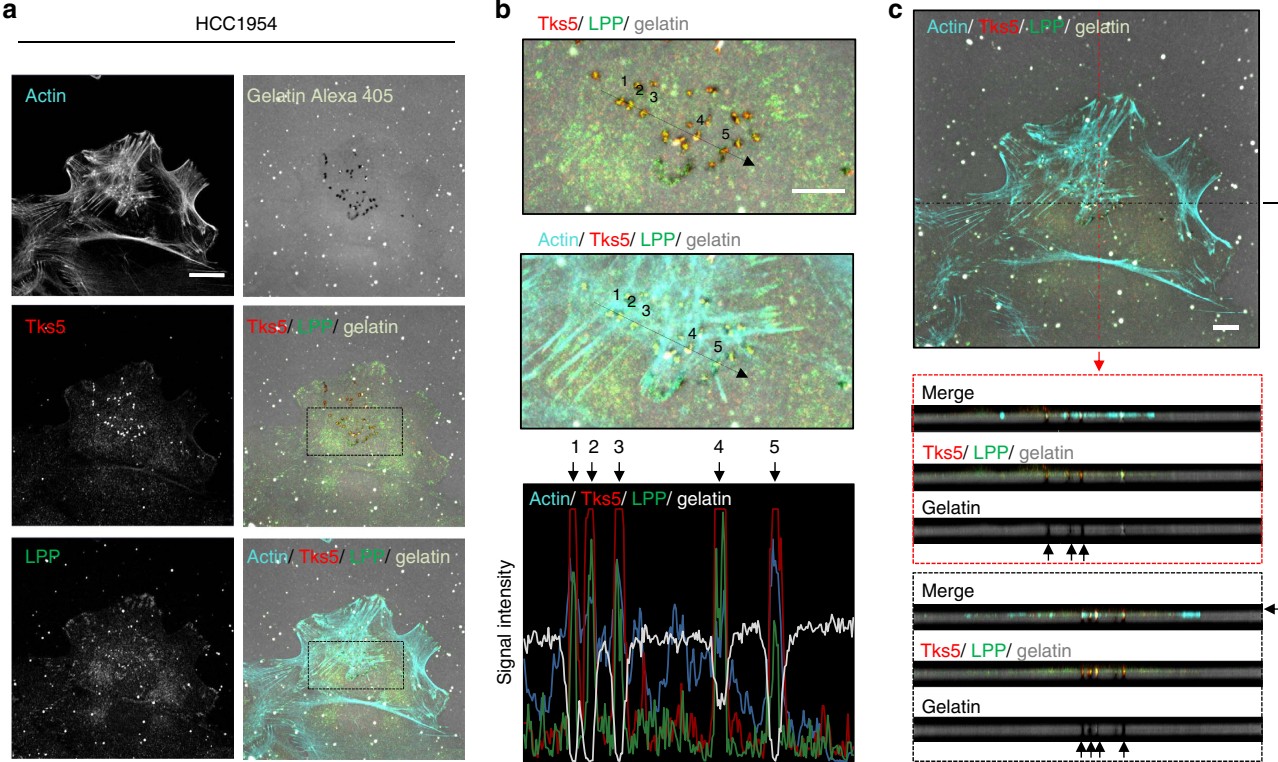

**Figure 4 | LPP co-localizes with Tks5 and actin at sites of ECM degradation.** HCC1954 breast cancer cells pre-treated with TGFβ (24 h) were plated onto Alexa 405-conjugated gelatin. Cells were fixed after 24 h and stained with antibodies against LPP, Tks5 and actin (phalloidin). (**a**) Confocal images were acquired to visualize gelatin degradation, and the localization of actin, Tks5 and LPP. Scale bar, 20 μm, and applies to all images. (**b**) Linescan analysis was performed over areas of degraded gelatin (indicated by black arrow) and the signal intensity of actin, Tks5, LPP and gelatin are shown for five regions of interest. Scale bar, 10 μm. (**c**) Z-stack acquisition was performed over 5.20 μm depth at 0.20 μm intervals. Orthogonal views (y–z plane: red box; x–z plane: black box) are presented. Black arrows indicate areas of gelatin degradation where LPP, Tks5 and actin are co-localized. Scale bar, 10 μm.

we sought to determine whether this loss of ECM degradation was due to an inability of cancer cells to upregulate or secrete MMPs. We observed that *MMP2*, *MMP9* and *MMP14* expression increased with TGFβ stimulation irrespective of LPP expression (Supplementary Fig. 7a). To address whether MMP activity is affected by TGFβ treatment, we also collected conditioned media (CM) from unstimulated and TGFβ-treated NMuMG-ErbB2 cells to assess MMP2 and MMP9 activity by gelatin zymography (Supplementary Fig. 7b). MMP2 and MMP9 activities were elevated across all NMuMG-ErbB2 cell populations, regardless of LPP expression, following TGFβ stimulation (3.5- and 2-fold, respectively; Supplementary Fig. 7b). These data argue that impaired invadopodia formation in breast cancer cells with diminished LPP expression does not result from defects in the proteolytic machinery associated with these structures.

**LPP-containing invadopodia promote cancer cell extravasation.** Our data demonstrate a role for LPP in breast cancer cell intravasation *in vivo* and invadopodia formation *in vitro*. Emerging data also reveals a critical role for invadopodia in augmenting cancer cell extravasation during metastasis[13,25]. Thus, we next investigated the potential role of LPP in promoting tumour cell extravasation using an *ex ovo* chick chorioallantoic membrane (CAM) assay. NMuMG-ErbB2 breast cancer cells harbouring LPP-shRNA and expressing eGFP-tagged LPP-WT, LPP-mLIM1 or LPP-ΔABD were intravenously injected into the CAM and monitored using high-resolution time-lapse intravital imaging. LPP co-localized with Tks5, an invadopodia-localized marker[8], in breast cancer cells expressing eGFP-LPP-WT, demonstrating that LPP is a constituent of invadopodia (Fig. 5a). Interestingly, cells with

eGFP-LPP-LIM1 exhibited discrete areas of LPP localization; however, Tks5 was no longer localized to cell protrusions (Fig. 5b). Finally, breast cancer cells expressing eGFP-LPP-ΔABD revealed a diffuse pattern of LPP and Tks5 expression (Fig. 5c). In agreement with our *in vitro* results, these observations demonstrate that breast cancer cells expressing eGFP-LPP-mLIM1 or eGFP-LPP-ΔABD fail to form functional invadopodia *in vivo*.

To assess the functional implications of these observations, the percentage of cells that extravasated out of the vasculature was determined. Expression of LPP-mLIM1 and LPP-ΔABD severely impaired the ability of breast cancer cells to extravasate, when compared to NMuMG-ErbB2 cells expressing wild-type LPP (LPP-WT) (Fig. 5d). We observed that eGFP-LPP-WT was localized to cellular protrusions that formed at the tumour cell/endothelial cell interface before the tumour cell breached the endothelial barrier. Moreover, once a tumour cell successfully traversed the vasculature at endothelial cell junctions, eGFP-LPP-WT was again localized to the tips of tumour cells that extended in the extravascular space (Supplementary Fig. 8a, upper panel; Supplementary Movie 1). In contrast, NMuMG-ErbB2 cells expressing either LPP-mLIM1 or LPP-ΔABD remain confined within the vascular network (Supplementary Fig. 8a, middle and lower panels, respectively; Supplementary Movie 1).

**LPP LIM1 and ABD motifs are required for lung metastasis.** We next assessed the effect of acutely reducing LPP expression on extravasation and lung metastasis formation following tail vein injection in mice. NMuMG-ErbB2 cells harbouring a doxycycline-inducible (dox) LPP-targeting shRNA (LPP-shRNA) were generated as previously described[17]. In agreement with the

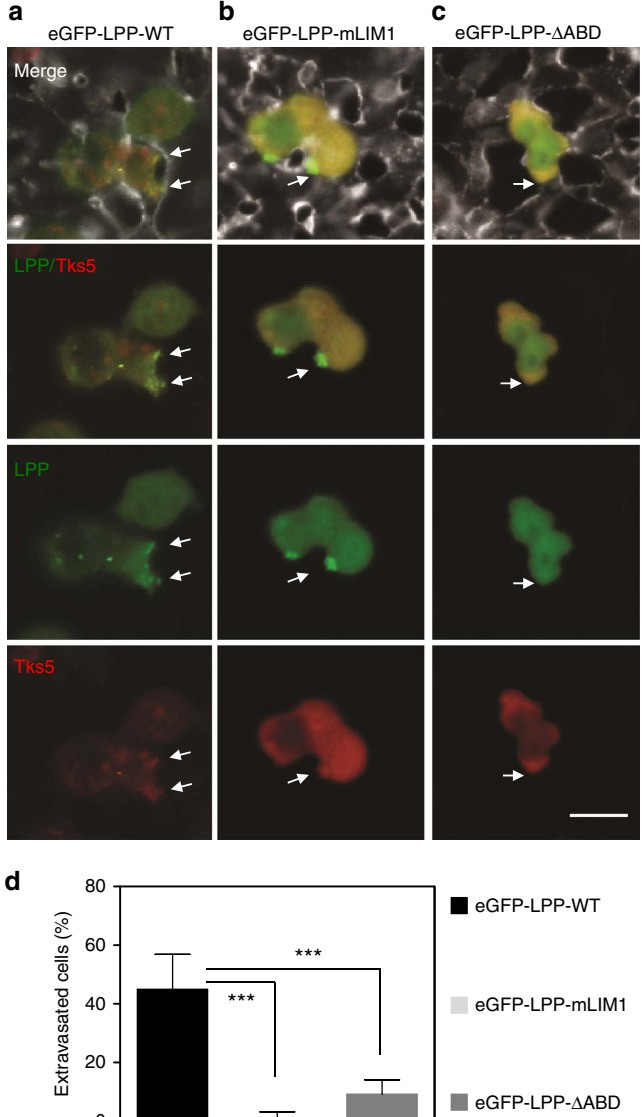

**Figure 5 | Wild-type LPP localizes to invadopodia and promotes *ex ovo* breast cancer cell extravasation.** NMuMG-ErbB2 LPP knockdown (LPP-shRNA) breast cancer cells harbouring a Tks5-mCherry fusion protein and expressing eGFP-LPP-WT (**a**), eGFP-LPP-mLIM1 (**b**) and eGFP-LPP-ΔABD (**c**) were intravenously injected into the CAM and monitored using high-resolution time-lapse intravital imaging in an *ex ovo* chick embryo model. White arrows point to regions where cancer cells contact the vascular wall. Chick endothelium is labelled with A647-lectin (grey signal). Scale bar, 20 μm, and applies to all images. (**d**) The percentage of breast cancer cells that have extravasated was quantified. The data represents quantification from ≥500 cells per cell line in *n*≥3 animals and error bars represent s.e.m. ***P<0.0001.

dispensable role for LPP in modifying primary tumour growth, NMuMG-ErbB2 cells expressing dox-inducible LPP shRNA formed mammary tumours that grew at similar rates when injected into doxycycline (LPP knockdown: LPP-shRNA, +Dox) or vehicle-treated (LPP proficient: LPP shRNA, −Dox) mice (Supplementary Fig. 9a). An acute reduction in LPP expression (+Dox) resulted in significantly impaired lung metastasis formation relative to controls (−Dox). Analysis of haematoxylin and eosin (H&E) stained lung sections revealed that doxycycline-treated mice (LPP knockdown) had fewer metastatic lesions than mice that did not receive doxycycline (LPP proficient)

(Supplementary Fig. 9b). The area of lung tissue occupied by lung metastases formed by LPP knockdown breast cancer cells (+Dox-treated mice) was also significantly smaller compared to the metastatic burden produced by LPP-proficient breast cancer cells (no Dox treatment) (Supplementary Fig. 9c). IHC staining confirmed that ErbB2 was similarly expressed in lung metastatic lesions formed by LPP proficient and LPP knockdown breast cancer cells (Supplementary Fig. 9d). Moreover, we observed that a significant diminishment in LPP expression was retained in lung metastases that emerged in Dox-treated (LPP knockdown) versus untreated mice (LPP proficient) (Supplementary Fig. 9d). These data demonstrate that LPP is important for efficient dissemination of ErbB2-expressing breast cancer cells to the lungs.

We next injected our panel of cell lines that were LPP proficient (eGFP-LPP-WT) or LPP deficient but rescued with LPP-mutants (eGFP-LPP-mLIM1 or eGFP-LPP-ΔABD) into the lateral tail vein of immunocompromised mice. LPP knockdown, or reconstitution with the eGFP-LPP-mLIM1 or eGFP-LPP-ΔABD mutants (Fig. 6a), severely impaired the formation of lung metastases compared to the high number of surface lesions evident on the lungs harvested from mice injected with NMuMG-ErbB2 cells that expressed endogenous LPP or were rescued with LPP-WT (Fig. 6b). To more carefully examine the lung metastatic burden in these cohorts, lungs were embedded in paraffin, sectioned and subjected to H&E staining (Fig. 6c,d). In agreement with quantification of the surface lesions, the number of metastases scored over four-step sections and the area of the lungs occupied by lung lesions (Fig. 6c) was greatly diminished in cohorts of mice injected with VC, LPP-mLIM1 and LPP-ΔABD expressing NMuMG-ErbB2 cells compared to LPP proficient and LPP-WT expressing NMuMG-ErbB2 cells. Together, our data demonstrate that breast cancer cell extravasation and lung metastasis require LPP and an intact LIM1 and ABD.

**LPP is a Src tyrosine kinase substrate**. Src is a constituent of invadopodia and it is required for the formation and function of these structures[8,26]. Interestingly, LPP has repeatedly been identified as a Src substrate in numerous proteomics-based screens[27–31]. In light of these observations, we assessed LPP tyrosine phosphorylation in a variety of breast cancer cell lines. LPP phosphorylation was elevated in NMuMG-ErbB2 cells following treatment with TGFβ (Fig. 7a). To investigate whether LPP tyrosine phosphorylation in response to TGFβ is mediated by Src family kinases (SFKs), we treated NMuMG-ErbB2 cells with two independent SFK inhibitors: Dasatinib and PP2. TGFβ-induced LPP tyrosine phosphorylation in NMuMG-ErbB2 cells was abrogated by treatment with either Dasatinib or PP2 (Fig. 7a). We extended our analyses to other ErbB2-expressing breast cancer cells (NIC and HCC1954) and a basal breast cancer cell line (BT549). In each of these breast cancer models, TGFβ stimulation induced LPP tyrosine phosphorylation, which was blunted by incubation with SFK inhibitors (Fig. 7a). Using the same lysates from which the immunoprecipitations were performed, we verified that treatment of these breast cancer cells with Dasatinib or PP2 effectively inhibited Src and FAK activation, while vehicle treatment (dimethylsulphoxide) had no effect (Supplementary Fig. 10).

To determine whether LPP is a *bona fide* Src target or whether the loss of tyrosine phosphorylation on LPP is due to off-target effects of SFK inhibitors, we expressed two independent Src-targeting shRNAs in NMuMG-ErbB2 cells (Fig. 7b). Total cell lysates from TGFβ-treated and untreated NMuMG-ErbB2 cells harbouring Src-shRNAs or an empty vector (EV) control

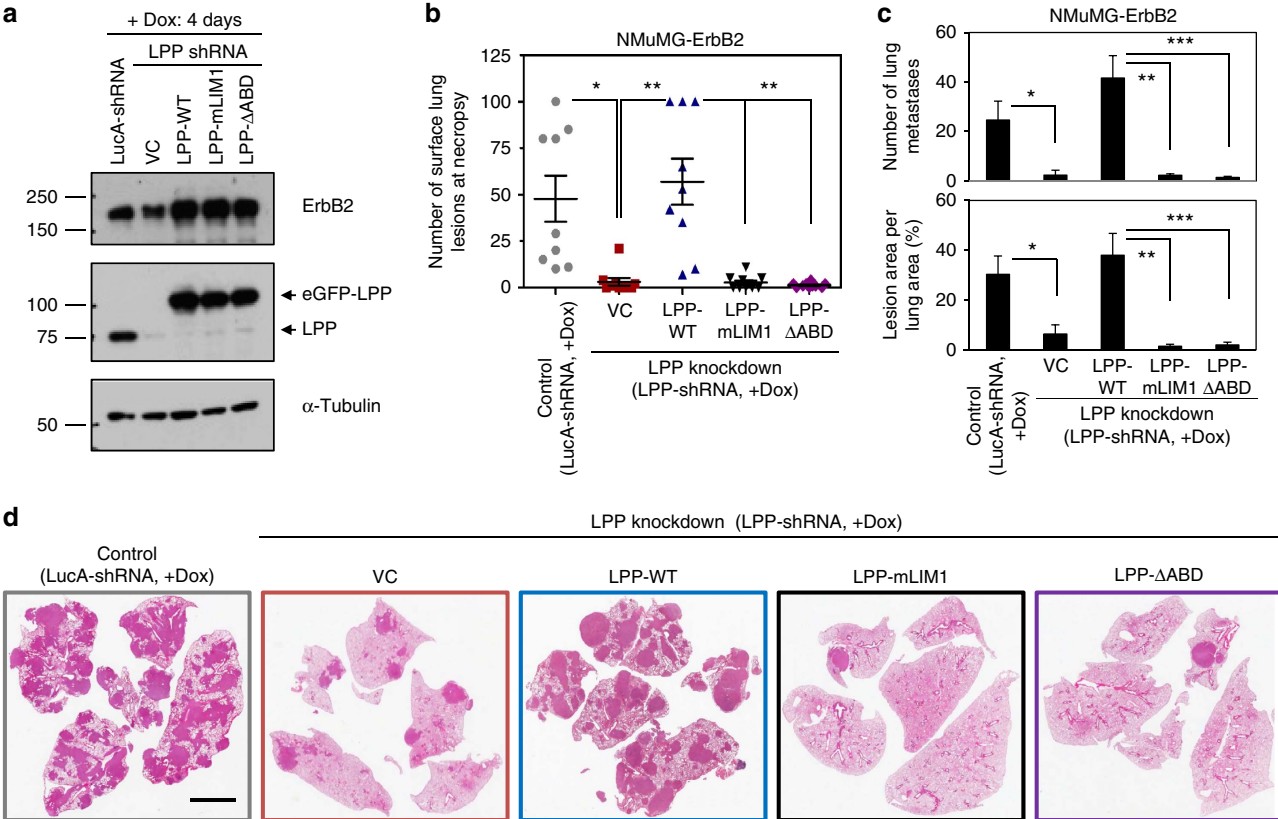

**Figure 6 | NMuMG-ErbB2 breast cancer cells expressing LPP mutants fail to efficiently establish breast cancer lung metastases.** NMuMG-ErbB2 breast cancer cells harbouring either a dox-inducible control shRNA against LucA or LPP-shRNA in which LPP rescue constructs were expressed (VC, LPP-WT, LPP-mLIM1, LPP-ΔABD), were employed in an experimental metastasis assay. The indicated breast cancer cells were injected into the tail vein of athymic mice ($n = 10$ per cohort). The breast cancer cells and the cohorts of mice were pre-treated with doxycycline starting 1 week before tumour cell injection and oral doxycycline was maintained throughout the experiment. (**a**) Immunoblot analysis was performed before breast cancer cell injection to assess ErbB2 and LPP expression levels. α-Tubulin was used as a loading control. (**b**) The number of macroscopic lesions on the lung surfaces was quantified at necropsy (*$P = 0.007$, **$P = 0.002$). (**c**) The number of lung metastases in each cohort were quantified from four H&E step sections (*$P = 0.038$, **$P = 0.038$, ***$P = 0.016$). The metastatic burden was determined from the same set of H&E samples and expressed as a percentage of the total lung tissue area (*$P = 0.005$, **$P = 7.7 \times 10^{-4}$, ***$P = 8.2 \times 10^{-4}$). Quantification was performed from 4-step sections, taken at 100 μm between each step from the following samples: LucA control: $n = 9$; VC: $n = 10$; LPP-WT: $n = 9$; LPP-mLIM1: $n = 9$; LPP-ΔABD: $n = 10$. (**d**) Representative H&E images are shown. Scale bar, 3 mm, and applies to all images in **d**. Error bars represent s.e.m. for all panels.

were immunoprecipitated for LPP and immunoblotted for phosphotyrosine. In line with our previous results, TGFβ induces LPP phosphorylation in NMuMG-ErbB2 cells with the EV; however, knockdown of Src impaired TGFβ-induced tyrosine phosphorylation of LPP (Fig. 7b). Control immunoblots revealed that Src and pSFK levels were significantly reduced in NMuMG-ErbB2 cells harbouring shRNAs targeting Src and that downstream FAK signalling was muted (Supplementary Fig. 11a). To determine if Src was sufficient to mediate LPP tyrosine phosphorylation, we expressed constitutively activated Src (Src-Y529F) in NMuMG-ErbB2 cells (Fig. 7c). In the presence of Src-Y529F, the basal level of LPP tyrosine phosphorylation was increased in the absence of TGFβ, which was further enhanced following TGFβ stimulation relative to control cells (Fig. 7c). Analysis of total cell lysates from NMuMG-ErbB2 cells expressing Src-Y529F revealed elevated levels of FAK phosphorylation on tyrosine residues known to be substrates for Src, thereby confirming enhanced Src-activity in these cells (Supplementary Fig. 11b). Finally, we examined the degree of LPP tyrosine phosphorylation in three independent NIC; Src[+/+] and NIC; Src[fl/fl] cell lines. Consistent with our previous data (Fig. 7a), LPP phosphorylation is induced in TGFβ-stimulated NIC breast cancer cells. In contrast, NIC cells that are Src-deficient (NIC; Src[fl/fl]) do not exhibit LPP tyrosine phosphorylation following TGFβ treatment (Fig. 7d and Supplementary Fig. 11c). Together, these data demonstrate that Src tyrosine kinase is both necessary and sufficient to mediate TGFβ-induced LPP tyrosine phosphorylation.

**Phospho-LPP is required for invasion and ECM degradation.** Proteomic studies focused on identifying Src substrates revealed five recurrently phosphorylated sites within LPP, which include: Y245, Y297/298 and Y301/302 (Fig. 8a)[27–30]. To assess the functional roles of these sites, all five tyrosine residues were simultaneously mutated to phenylalanine (F), creating a LPP-5F mutant with an amino-terminal eGFP fusion (Fig. 8a). We stably reduced endogenous LPP levels (LPP-shRNA) in NMuMG-ErbB2 cells and performed rescue experiments in which we expressed either eGFP-LPP-WT or eGFP-LPP-5F in these cells (Fig. 8b). The complete panel of NMuMG-ErbB2 cells was subjected to both migration and invasion assays using Boyden chambers. In agreement with our previous results[17], knockdown of LPP completely abrogated TGFβ-induced cell migration and invasion, which is rescued with the expression of exogenous eGFP-LPP-WT (Fig. 8c,d). Surprisingly, TGFβ treatment resulted in a two-fold increase in the migration of cells expressing eGFP-LPP-5F, which

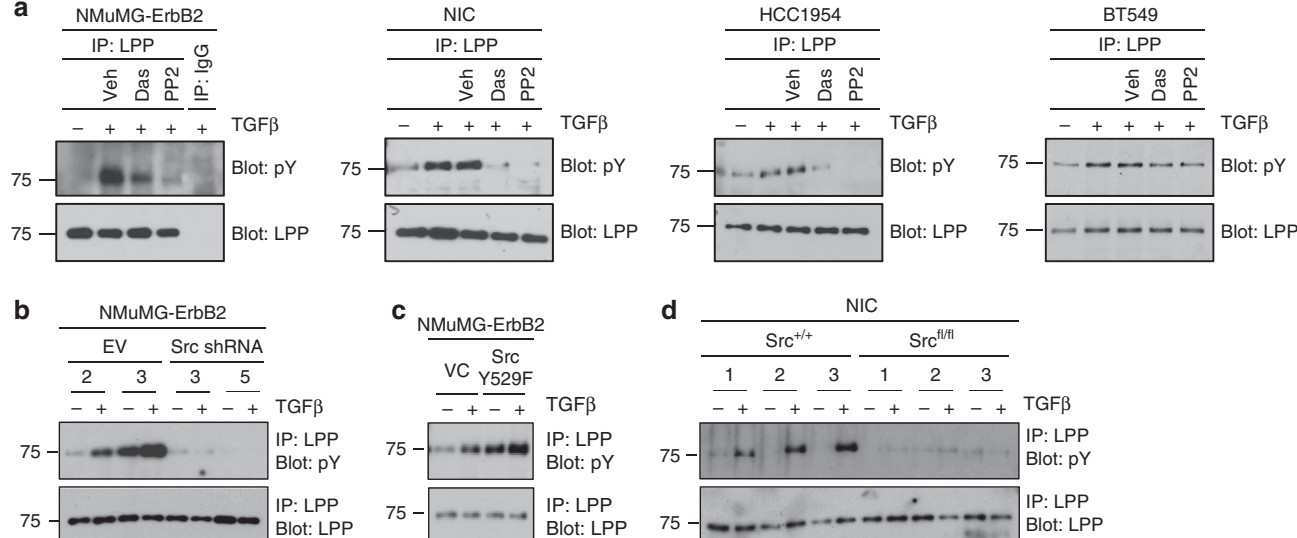

**Figure 7 | TGFβ-induced LPP phosphorylation is Src dependent. (a)** NMuMG-ErbB2, NIC, HCC1954 and BT549 breast cancer cell lines were stimulated with or without TGFβ, and incubated with inhibitors against Src Family Kinases (Dasatinib and PP2) or vehicle control (dimethylsulphoxide). Total cell lysates were immunoprecipitated with antibodies against LPP and immunoblotted for phosphotyrosine (pY) or LPP. **(b)** NMuMG-ErbB2 cells were infected with two independent Src-shRNAs, along with EV controls. Following 24 h of stimulation with or without TGFβ, LPP immunoprecipitates were immunoblotted for phosphotyrosine (pY) and LPP. **(c)** NMuMG-ErbB2 cells were transfected with a construct encoding a constitutively activated Src kinase and treated with or without TGFβ (24 h). Tyrosine phosphorylation (pY) and LPP levels were assessed by immunoblot from LPP immunoprecipitates. **(d)** Total cell lysates from three independent NIC; Src$^{+/+}$ and three independent NIC; Src$^{fl/fl}$ breast cancer cell lines were immunoprecipitated for LPP following treatment with or without TGFβ (48 h). LPP tyrosine phosphorylation (pY) and total LPP levels were assessed by immunoblot.

was comparable to cells expressing eGFP-LPP-WT (1.80-fold change; Fig. 8c). In contrast, expression of eGFP-LPP-5F failed to restore the TGFβ-induced invasion of NMuMG-ErbB2 cells that was observed following eGFP-LPP-WT expression (1.76-fold change; Fig. 8d). These data argue that tyrosine phosphorylation of LPP is critical for breast cancer cell invasion, but dispensable for migration, in response to TGFβ.

To assess whether tyrosine phosphorylation of LPP is important for invadopodia formation and ECM degradation, TGFβ-treated NMuMG-ErbB2 cells were plated on fluorescently labelled gelatin substrate for 24 h and the area of invadopodia-mediated gelatin degradation, as observed by a loss of fluorescent signal, was subsequently quantified (Fig. 8e). NMuMG-ErbB2 cells harbouring LucA-shRNA readily degraded the gelatin substrate, while reduced LPP expression (LPP-shRNA) abrogated the degradative ability of these cells. Although expression of eGFP-LPP-WT rescued the proteolytic phenotype of NMuMG-ErbB2 cells, eGFP-LPP-5F expressing cells were unable to induce gelatin degradation (Fig. 8e). Together, these data functionally demonstrate that tyrosine phosphorylation of LPP is critical for invadopodia formation.

**Phospho-LPP (Y245/301/302) mediates cancer cell invasion.** To define the precise tyrosine residues within LPP required for TGFβ-induced invasion of NMuMG-ErbB2 breast cancer cells, we constructed a panel of LPP mutants in which tyrosine (Y) residues at positions 245, 297/298 and 301/302 were mutated to phenylalanine residues (F), either independently or in combination (Fig. 9a). Equivalent expression levels of each LPP construct were determined by immunoblot analysis (Fig. 9b). We subjected the panel of NMuMG-ErbB2/LPP-shRNA cells expressing eGFP-LPP phospho-mutants, along with control NMuMG-ErbB2/Luc-shRNA cells, to migration and invasion assays. Consistent with results obtained using the LPP-5F mutant (Fig. 8c), individual expression of each LPP tyrosine

phosphorylation mutant supported TGFβ-induced cell migration (Fig. 9c, upper graph). Similarly, LPP-shRNA cells expressing LPP-245F (Y245F), LPP-A (Y297/298F), LPP-B (Y301/302F) and LPP-C (Y245/297/298F) rescue constructs also supported increased cell invasion in response to TGFβ stimulation (Fig. 9c, lower graph). Strikingly, LPP-shRNA cells expressing LPP-D (Y245/301/302F) rescue construct failed to exhibit TGFβ-induced cell invasion (Fig. 9c, lower graph). These results argue that tyrosine residues Y245, Y301 and Y302 within LPP cooperate to promote breast cancer cell invasion in response to TGFβ.

**Phospho-LPP (Y245/301/302) promotes breast cancer metastasis.** To assess whether loss of LPP phosphorylation on these sites also abrogates breast cancer metastasis, we injected NMuMG-ErbB2 cells expressing a panel of LPP phospho-mutants (Fig. 9a), along with LucA-shRNA control and LPP-shRNA cells into the mammary fat pads of immunocompromised mice (Fig. 10). Consistent with our previous results (Fig. 1 and Supplementary Fig. 4), we did not observe any significant differences in tumour volume (Fig. 10a) or tumour burden at end point (Fig. 10b) between LucA-shRNA control, LPP-shRNA, LPP-WT, LPP-245F, LPP-C and LPP-D tumour-bearing animals.

We collected whole blood from these animals by cardiac puncture to isolate circulating tumour cells (CTC). In line with our previous results (Fig. 2), loss of LPP significantly reduced the number of CTCs, when compared to LucA-shRNA control (Fig. 10c). The expression of LPP-WT, LPP-245F and LPP-C was sufficient to promote the intravasation of NMuMG-ErbB2 breast cancer cells, while the expression of LPP-D reduced the average number of detectable CTCs.

To assess the metastatic ability of NMuMG-ErbB2 breast cancer cells expressing different LPP phospho-mutants, we collected lung tissues at necropsy and quantified the number of macroscopic metastatic lesions (Fig. 10d, upper graph). Loss of LPP significantly reduced the number of metastatic lesions when compared

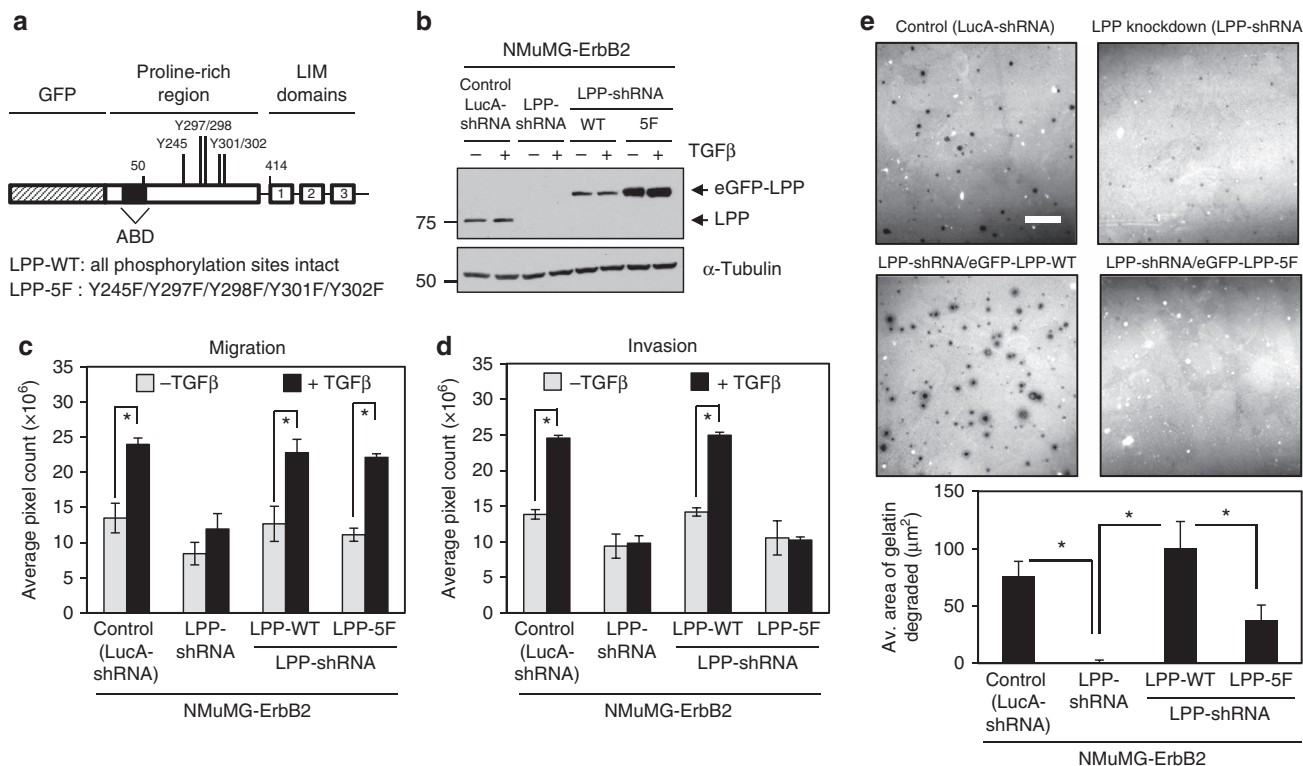

**Figure 8 | Tyrosine phosphorylation of LPP is required for TGFβ-induced breast cancer cell invasion and invadopodia formation but is dispensable for TGFβ-induced cell migration.** (**a**) Schematic diagram of eGFP-tagged LPP constructs. LPP is composed of a PRR (amino acids 1–413) and three LIM domains (amino acids 414–613). An ABD within LPP is located between residues 41–57 (indicated by black box). The positions of five tyrosine (Y) residues that were mutated to phenylalanine (F) residues within eGFP-LPP-5F are indicated. (**b**) Immunoblot analyses of LPP levels in a panel of NMuMG-ErbB2 cells treated with or without TGFβ (LucA-shRNA: expressing non-targeting shRNA against LucA, LPP-shRNA: expressing shRNA against the 3′-UTR of LPP, in which eGFP-LPP-WT or eGFP-LPP-5F were expressed). α-Tubulin was used as a loading control. (**c**,**d**) NMuMG-ErbB2 cell populations, treated with or without TGFβ, were subjected to migration and invasion assays. The data is expressed as the average pixel count obtained from four independent migration and three independent invasion experiments performed in duplicate (*$P < 0.004$) and error bars represent s.e.m. (**e**) NMuMG-ErbB2 cells harbouring either a shRNA against LucA (LucA-shRNA) or targeting LPP (LPP-shRNA) were used to express eGFP-LPP-WT and eGFP-LPP-5F rescue constructs. The indicated panel of NMuMG-ErbB2 cells were pre-treated with TGFβ (24 h), and subsequently plated onto fluorescently labelled gelatin for an additional 24 h. The area of degradation was quantified from images of LucA-shRNA control ($n = 26$), LPP-shRNA ($n = 11$), eGFP-LPP-WT ($n = 14$) and eGFP-LPP-5F ($n = 15$) cells from four independent experiments, and error bars represent s.e.m. Scale bar, 20 μm, and applies to all images in **e**. (*$P < 0.01$).

to control cells expressing endogenous LPP. Re-expression of LPP-WT was able to fully restore the metastatic phenotype, while mutation of LPP tyrosine residue 245 (LPP-245F) or tyrosine residues 245/297/298 (LPP-C) to phenylalanine residues did not affect the metastatic ability of breast cancer cells. In contrast, simultaneous mutation of tyrosine residues 245/301/302 (LPP-D) significantly decreased the number of surface lung metastatic lesions, albeit not to the same extent as the loss of LPP. Further analysis of H&E lung sections to determine the area of metastatic burden revealed that the loss of LPP significantly reduces the size of metastatic lesions (LucA-shRNA versus LPP-shRNA) but the metastatic ability of breast cancer cells can be rescued by the expression of LPP-WT, LPP-245F and LPP-C. However, expression of LPP-D failed to restore the metastatic phenotype (Fig. 10d, lower graph and Fig. 10e). Collectively, these data reveal that breast cancer intravasation and metastasis formation requires phosphorylation of LPP on tyrosine residues 245, 301 and 302.

## Discussion

The present study is the first, to our knowledge, to demonstrate that LPP is dispensable for primary mammary tumour growth but promotes metastasis of ErbB2-expressing breast cancer cells. Our data implicate LPP as a component of invadopodia required

for the formation of these structures following TGFβ stimulation. Increasing evidence demonstrates that TGFβ signalling can enhance podosome formation in non-transformed cells and invadopodia formation in cancer cells[15,32,33]. Invadopodia are crucial cellular structures that allow breast cancer cells to escape the primary tumour by degrading ECM components, enabling cancer cell intravasation into the circulatory system and extravasation into secondary organs[12,13,25,34]. Mounting evidence suggests a direct correlation between the propensity of cancer cells to form invadopodia and poor prognosis in breast cancer patients. Consequently, these structures are emerging as a promising target to impair metastasis formation[23,24,35]. Our data implicate LPP as regulator of invadopodia formation and function, performing critical roles during breast cancer cell intravasation and extravasation during the metastatic process. Loss of LPP does not impair TGFβ-induced MMP expression (*MMP2*, *MMP9* or *MMP14*) or TGFβ-induced MMP activity (MMP2 and MMP9). Thus, the failure of breast cancer cells to form invadopodia under low LPP expression does not appear to result from impaired proteolytic activity of these structures.

Our results in both human and mouse ErbB2-expressing breast cancer cells reveal that LPP co-localizes with Tks5 and actin at sites of matrix degradation, revealing that a population of

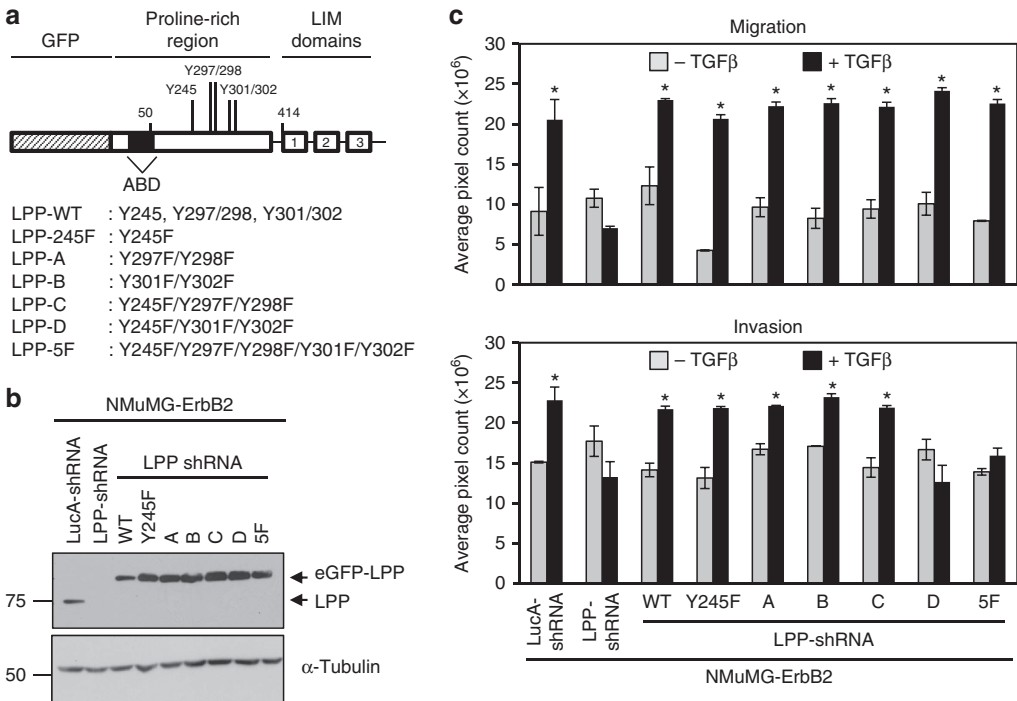

**Figure 9 | LPP phosphotyrosine sites at residues 245/301/302 are required for TGFβ-induced cell invasion. (a)** Schematic diagram of eGFP-tagged LPP constructs indicating the location of the tyrosine (Y) residues that were converted to phenylalanine (F) residues to create a panel of LPP phospho-mutants. **(b)** Immunoblot analyses of LPP levels in a panel of NMuMG-ErbB2 cells (control: LucA-shRNA; LPP knockdown: LPP-shRNA; LPP rescues: cells expressing LPP-shRNA, in which eGFP-LPP-WT or individual eGFP-LPP phospho-mutants were expressed). α-Tubulin was used as a loading control. **(c)** NMuMG-ErbB2 cell populations, treated with or without TGFβ, were subjected to migration and invasion assays. The data are expressed as the average pixel count obtained from three independent experiments performed in duplicate, and error bars represent s.e.m. (*$P < 0.0001$).

cellular LPP is localized to invadopodia. Using the *ex ovo* CAM assay, we were able to visualize the location of LPP during breast cancer cell extravasation. Our data show that WT LPP co-localizes with Tks5, supporting the idea that LPP is a component of invadopodia that form during extravasation. In contrast, the LPP-ΔABD mutant fails to localize to discrete structures and Tks5 expression appears diffuse. More striking is the observation that a LPP mutant that cannot target to focal adhesions (LPP-mLIM1) still localizes to discrete regions at the tips of cellular protrusions, but Tks5 exhibits a diffuse expression pattern. LPP initially localizes to the breast cancer/endothelial cell interface, precisely where the tumour cell breaches the endothelial barrier. Once tumour cells have begun the trans-endothelial migration process, LPP-containing structures can also be detected at the protrusive tips of the cancer cells. The phenotypes associated with each of these LPP constructs suggests that LPP may facilitate invadopodia formation and function by engaging the actin cytoskeleton and contributing to the protrusive force that is required to form the degradative protrusions that eventually help cancer cells breach the basement membrane beneath the endothelial cell barrier. Indeed, it is now emerging that invadopodia are mechanosensitive structures[36,37]. Essential roles for actin-bundling proteins within invadopodia have been previously established; Fascin, for example, has been shown to stabilize the actin core and generate protrusive force[38]. Interestingly, both α-actinin and vasodilator-stimulated phosphoprotein (VASP) localize along the shaft of mature invadopodia and are required for their elongation[39]. Together, these data support a model in which LPP localizes to invadopodia, engages α-actinin to generate actin cross-links necessary for force generation within these ventral protrusions.

Previous proteomics studies have indicated that LPP may be a substrate of Src[27–31]. However, our results also functionally implicate Src-mediated LPP tyrosine phosphorylation as an important determinant of its ability to promote breast cancer invasion. Key LPP tyrosine phosphorylation sites are situated within the PRR, between the α-actinin binding domain (amino acids 41–57) and the LIM domains (amino acids 414–613). Interestingly, these same key residues are hyper-phosphorylated in PTP1B-deficient fibroblasts[40] and several proteomics studies have since identified LPP as a PTP1B interactor[40,41]. Thus, it is conceivable that elevated PTP1B may be important to de-phosphorylate LPP, which would permit invadopodia disassembly. Conversely, PTP1B can activate Src through dephosphorylation of Y529 (ref. 42), which may lead to elevated LPP phosphorylation and enhanced invadopodia formation[43]. In this regard, we show that a constitutively active Src (Src-Y529F) can enhance LPP tyrosine phosphorylation. Indeed, genetic ablation of PTP1B delays ErbB2 breast tumour progression and metastasis[44]. Thus, it is tempting to speculate that Src and PTP1B coordinately regulate the phosphorylation/dephosphorylation cycle of LPP, respectively, to enhance cellular invasion and breast cancer metastasis.

Our data clearly show that LPP-mediated cellular invasion can be uncoupled from the role of LPP in cell migration. Diminished expression of LPP causes defects in both TGFβ-induced cell migration and invasion; however, tyrosine phosphorylation on LPP (Y245/301/302) specifically regulates cell invasion and metastasis. It is conceivable that phosphorylation on LPP results in conformational changes that permit access to additional binding partners that mediate cellular invasion/metastasis. Indeed, LPP plays a dual role in contractile smooth muscle cells (SMC). In mature smooth muscle tissues, LPP localizes at

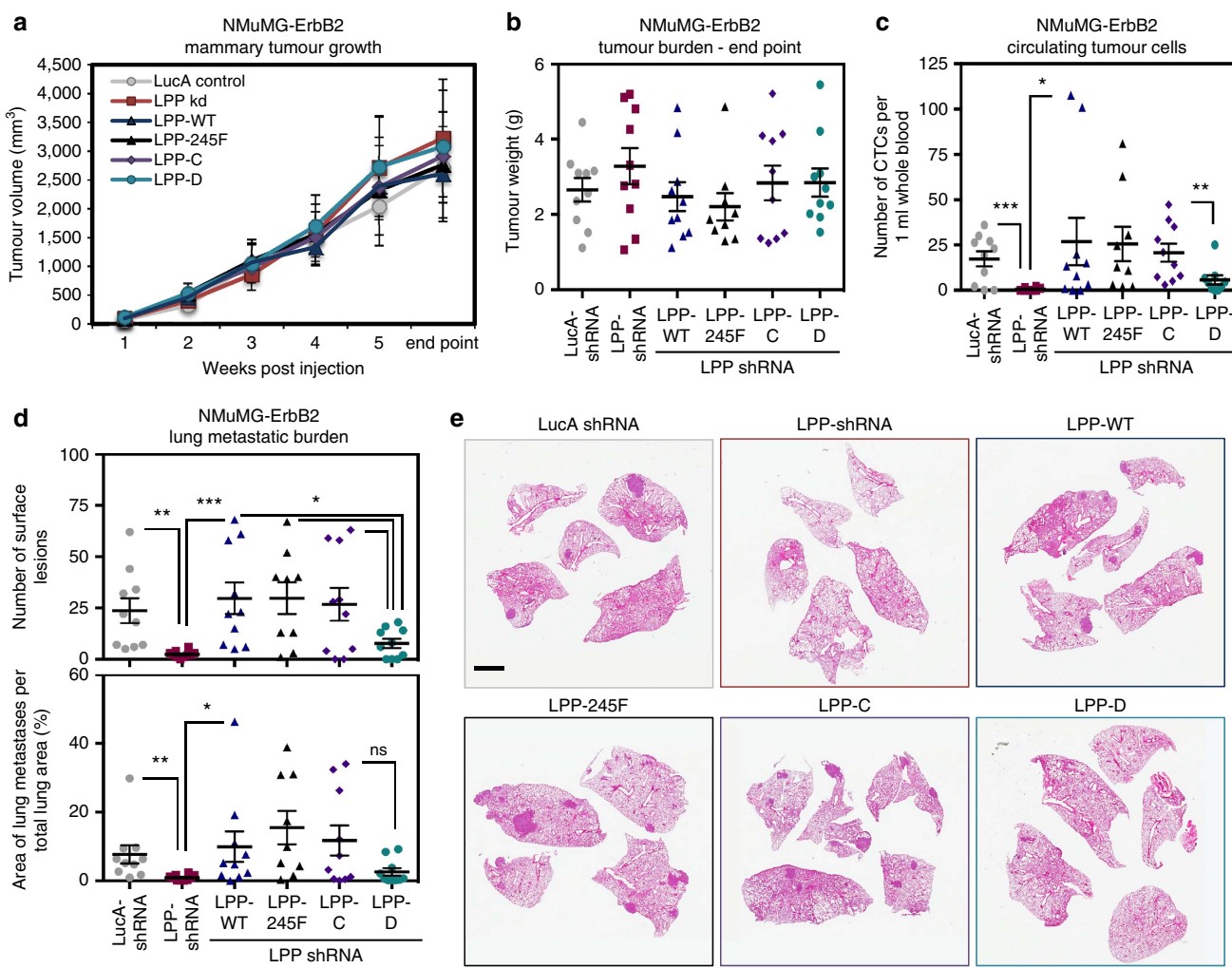

**Figure 10 | Phosphorylation on LPP tyrosine residues 245/301/302 is required for efficient breast cancer lung metastasis.** NMuMG-ErbB2 breast cancer cells expressing LucA-shRNA and LPP-shRNA with LPP rescue constructs (LPP-WT, LPP-245F, LPP-C and LPP-D) were injected into the mammary fat pads of athymic mice ($n = 10$ per cohort). (**a**) Mammary tumour growth was determined by weekly caliper measurements. (**b**) At the time of euthanasia, tumours were collected and weighed to determine the final tumour burden. (**c**) Whole blood was collected by cardiac puncture and the number of CTC-derived adherent colonies was assessed 2 weeks post isolation (***$P = 0.003$, **$P = 0.02$, *$P = 0.05$). (**d**) Lungs were collected at necropsy and the number of macroscopic surface lesions was quantified (***$P = 0.006$, **$P = 0.007$, *$P \leq 0.04$). The area of metastatic burden was quantified from four H&E stained lung sections per animal, and is expressed as a percentage of total lung surface area (*$P = 0.03$, **$P = 0.04$, ns; $P = 0.07$) (**e**) Representative H&E lung sections are shown. Scale bar, 2 mm and applies to all images in **e**. Error bars represent s.e.m. for all panels.

peripheral dense bodies, which are sites of actin filament insertion in contractile SMCs[45], and these structures are enriched in α-actinin[46–48]. On the other hand, LPP can also localize to vinculin-based focal adhesions in migrating SMCs. Together, these data suggest that LPP can play unique roles within different cellular structures (focal adhesions versus invadopodia), depending on the context. In agreement with our work, LPP promotes cell migration through localization to focal adhesions; however, our data also reveal that LPP localizes to invadopodia to promote cellular invasion in response to phosphorylation by Src. Nonetheless, the precise mechanisms underlying LPP phosphorylation-dependent breast cancer cell invasion remain to be elucidated.

We have previously demonstrated that LPP localizes to focal adhesions in response to TGFβ and this localization is critical for migration and invasion of breast cancer cells[17]. In this current study, we show that LPP can also localize to and functionally regulate invadopodia activity, whereas a mutant that cannot localize to focal adhesions (LPP-mLIM1) fails to form such

invasive structures. While many of the proteins that comprise focal adhesions and invadopodia are shared, it is unclear whether invadopodia are derived from focal adhesions or they are discreet structures that assemble independently of one another[49,50]. Indeed, several proteins have been found to localize to both focal adhesions and invadopodia, raising the idea that focal adhesions and invadopodia are intimately linked. One possibility is that focal adhesion proteins are recruited into adhesional rings to help anchor invadopodia formation at early stages[15,51,52]. It is conceivable that biochemical (ECM composition) or physical cues (substrate stiffness, tractional forces) may switch LPP from a pro-migratory role in focal adhesions to a pro-invasive function within invadopodia. Indeed, it has been shown that in non-transformed cells, podosomes rather than focal adhesions form at sites of integrin-matrix attachments in the absence of tractional forces[53]. Furthermore, emerging evidence now suggests that cancer cells will migrate quickly on stiff substrates[54] and form invasive-like protrusions on softer matrices[55]. However, the

influence of tissue stiffness on invadopodia formation is contradictory, with some studies indicating that stiffer matrices also support more invadopodia and increased cellular invasion[56–58]. Whether mechanical cues can direct LPP to focal adhesions to promote cellular migration or to invadopodia to enhance breast cancer invasion awaits further investigation.

## Methods

**Cell culture.** NMuMG parental cells[20] and HCC1954 cell line were obtained from the ATCC. NIC cells were derived from ErbB2-expressing mammary tumours that arose in transgenic mice[22]. The NMuMG-ErbB2 cell population, the NIC cell populations, and the HCC1954 cells were maintained in their appropriate media[17]. Where indicated, NMuMG-ErbB2 cells were treated with $1\,\mu g\,mL^{-1}$ doxycycline (Cat. #: D9891, Sigma-Aldrich) for at least 96 h before further experimentation. All cell populations were stimulated with $2\,ng\,ml^{-1}$ of TGFβ1 (Cat. #: 100-21, Peprotech) for 24–48 h, as indicated. Mycoplasma screening was routinely performed using MycoAlert mycoplasma detection kit (Cat. #: LT07-318, Lonza).

NIC; Src$^{+/+}$ and NIC; Src$^{fl/fl}$ cell lines were established from MMTV-NIC tumours homozygous for a conditional c-Src allele[59,60]. Briefly, mammary tumours were excised and processed using a McIlwain tissue chopper (Mickle Laboratory Engineering) and dissociated for 2 h in $2.4\,mg\,ml^{-1}$ each of Collagenase B and Dispase II dissolved in DMEM. Dissociated cells were washed three times with 1 mM EDTA in PBS and plated in DMEM supplemented with 5% FBS, $5\,\mu g\,ml^{-1}$ insulin, $1\,\mu g\,ml^{-1}$ hydrocortisone, $5\,ng\,ml^{-1}$ EGF and $35\,\mu g\,ml^{-1}$ Bovine Pituitary Extract.

Retroviral production of LucA-shRNA, LPP-shRNA and eGFP-LPP rescue constructs was performed using Retro-X Universal Packaging System (Cat. #: 631530, Clontech). NMuMG-ErbB2 and NIC cells were then incubated with polybrene ($8\,\mu g\,ml^{-1}$) and virus containing media for 24 h to allow for retroviral infection.

**DNA constructs and siRNA.** The doxycycline inducible-shRNA constructs against LucA, as a control, and against LPP were introduced into the tetracycline-regulated retroviral vector-SIN-TREmiR30-PIG (TMP) vector system (Cat. #: EAV4678, Open Biosystems)[17]. Stable LPP knockdown and control constructs were generated by restriction enzyme digestion of the TMP-shRNA LucA and TMP-shRNA LPP mentioned above. The resulting shRNA fragments were inserted into XhoI/EcoRI sites of the MSCV-LTRmiR30-PIG (LMP) vector system (Cat. #: EAV4071, Open Biosystems). The sequences within the TMP and LMP vectors encoding GFP were removed by restriction enzyme digestion and all constructs were introduced into the cells via retroviral infection. NMuMG-ErbB2 cells expressing TMP-shRNA or LMP-shRNA against LPP were rescued with pMSCV-eGFP as an EV control (VC) or with the following pMSCV eGFP-tagged LPP constructs: wild-type LPP (LPP-WT), LPP with LIM1 domain mutations (LPP-mLIM1), LPP with ABD deletion (LPP-ΔABD) or with LPP phospho-mutants as indicated. Generation of the LPP mutants was performed with QuikChange site directed mutagenesis kit (Cat. #: 210518, Stratagene) according to manufacturer's protocol[17].

HCC1954 cells were transfected with the following siRNA sequences against LPP: GGA-AGA-UAG-UCU-UAU-GUA, CCC-AGU-UUA-AGA-CAC-CAA and GCC-AAG-UUA-AAU-AGC-AAA (Cat. #: HSC.RNAI.N005578.12.1, HSC.RNAI.N005578.12.2, HSC.RNAI.N005578.12.3, Integrated DNA Technologies). Cells were transfected with a pool of siRNAs, each at a concentration of 2 nM, or a control non-targeting siRNA (AGU-UAA-UCG-CGU-AUA-AUA) using INTERFERrin transfection reagent (Cat. #: 409-50, Polyplus transfection) according to manufacturer's protocol. To confirm efficient LPP knockdown, total cell lysates were collected at the end point of each gelatin degradation assay and immunoblotted for LPP.

The pUSE src-529F construct was purchased from Upstate Biotechnology (Cat. # 21–115, Upstate Biotechnology). Lentiviral shRNA vectors against Src (src-shRNA3: TRCN0000023596; src-shRNA5: TRCN0000278660) were obtained and prepared as described elsewhere[61].

**RNA isolation and real-time quantitative reverse transcription PCR.** Cells incubated in the presence or absence of TGFβ for 24 or 48 h were subjected to RNA isolation using QIAshredder columns (Cat. #: 79656, Qiagen) and RNeasy Mini Kit (Cat. #: 74106, Qiagen) as per manufacturer's instructions. Reverse transcription was performed with $1\,\mu g$ of total RNA using a high-capacity cDNA reverse transcription kit (Cat. #: 4368813, Applied Biosystems). All cDNA samples obtained from Reverse Transcriptase PCR were then diluted 1:50 in ddH$_2$O. Diluted cDNA ($5\,\mu l$), 667 nM of each forward and reverse primer, and FastStart Universal SYBR Green Master (Cat. #: 04913914001, Roche) were subjected to quantitative real time PCR analysis in Rotor-gene RG-3000 PCR system (Corbett Research). The normalized expression level of the target genes was calculated as follows, using the average threshold cycle (CT) value from duplicate measurements:

$$Efficiency_{target} + 1^{\Delta CT target\,(control-sample)}/Efficiency_{reference} + 1^{\Delta CT reference\,(control-sample)}$$

The data shown is the average expression level normalized to mouse *GAPDH* and presented the relative fold change to the calibrator (unstimulated LucA-shRNA control) from three independent experiments. The following primer sequences were used: *MMP2* (Forward: 5′-CAAGTTC-CCCGGCGATGTC-3′/Reverse: 5′-TTCTGGTCAAGGTCACCTGTC-3′); *MMP9* (Forward: 5′-TCGCGTGGATA AGGAGTTCT-3′/Reverse: 5′-CGGTTGAAGCAAAGAAGGAG-3′); *MMP14* (Forward: 5′-CAGTATGGCTACCTACCTCCAG-3′/ Reverse: 5′-GCCTTGCC TG-TCACTTGTAAA-3′); *GAPDH* (Forward: 5′-CAAGTATGATGACATCAA G-AAGGTGG-3′/Reverse: 5′-GGAAGAGTGGGAGTTGCTGTTG-3′). MMP2 and MMP14 sequences were obtained from PrimerBank (http://pga.mgh.harvar-d.edu/primerbank/).

**Proliferation assays.** NMuMG-ErbB2 cells ($5\times10^3$) and $1\times10^4$ NIC cells were plated into xCELLigence E-plates (Cat. #: 05469813001, Roche Applied Science) and incubated in the presence or absence of TGFβ. Cell growth was monitored in a RTCA DP Analyzer (Roche Applied Science) for up to 96 h and the doubling time was calculated using the xCELLigence RTCA software (Roche Applied Science) according to manufacturer's protocol.

**Immunoblotting.** Total cell lysates ($20\,\mu g$) were resolved by 6–12% SDS–polyacrylamide gel electrophoresis. Proteins were transferred onto polyvinylidene difluoride membranes (Cat. #: IPVH00010, Millipore) and membranes were blocked in 5% fat-free milk or bovine serum albumin. Membranes were incubated with the following antibodies: LPP (1:1,000; Cat. #: 3389, Cell Signaling), ErbB2 (1:200; Cat. #: SC-284, Santa Cruz), E-cadherin (1:1,000; Cat. #: U3254, Sigma), Fibronectin (1:5,000; Cat. #: F3648, Sigma), Vimentin (1:1,000; Cat. #: 550513, BD Biosciences), α-SMA (1:1,000; Cat. #: A2547, Sigma), pSmad2 (1:1,000; Cat. #: 3101, Cell Signaling), Smad2/3 (1:1,000; Cat. #: 3102, Cell Signaling), Snail (1:1,000; Cat. #: 3895, Cell Signaling), 4G10 platinum (1:1,000; Cat. #: 05-1050, Millipore), Phospho-src Family Tyr416 (1:1,000; Cat. #: 2101 S, Cell Signaling), Src (1:1,000 clone GD11; Cat. #: 05-184, Millipore), Phospho-FAK Tyr397 (1:1,000; Cat. #: 3283, Cell Signaling), Phospho-FAK Tyr576/577 (1:1,000; Cat. #: 3281, Cell Signaling), Phospho-FAK Tyr925 (1:1,000; Cat. #: sc-11766 R, Santa Cruz), FAK (1:1,000; Cat. #: 06-543, Millipore) and α-Tubulin (1:40,000; Cat #: T9026, Sigma). The appropriate HRP-conjugated secondary antibodies (1:10,000; Jackson Immuno Research Laboratories) were incubated for 1 h and the membranes were visualized using Pierce Enhanced Chemiluminescence (ECL) (Cat #: 32106, Thermo Scientific). Uncropped scans of all immunoblots in the main figures of this article can be found in the Supplementary Information (Supplementary Fig. 13).

**Gelatin degradation assay and immunofluorescence staining.** To label Alexa 405-conjugated gelatin[62], Alexa 405 dye (Cat. #: A30000, ThermoFisher Scientific) was resuspended in sodium bicarbonate, and incubated with 0.2% Gelatin (Cat. #: GEL771, Bioshop) for 1 h. The solution was then passed through Bio-Gel P-30 (Cat. #: 150-4154, Bio-Rad) to separate out unlabelled dye. A ultraviolet lamp was used to visualize the fractions within the column. The eluted Alexa 405 gelatin fraction was stored in the dark at $4\,^{\circ}C$ until ready to use.

Gelatin-degradation assays were performed on fluorescently conjugated gelatin-coated coverslips[63]. Briefly, sterile coverslips were coated with a mix of $0.1\,mg\,ml^{-1}$ poly-D-lysine (Cat. #: P6407, Sigma) and $5\,\mu g\,(cm^2)^{-1}$ Fibronectin (Cat. #: FC010, Millipore) in PBS for 20 min, followed by incubation with 0.4% Glutaraldehyde for 20 min. Oregon Green 488 conjugated gelatin (Cat. #: G13186, Invitrogen) or in-house made Alexa 405 gelatin, was diluted by 1:50 with 0.1% unconjugated gelatin (Cat. #: 07903, Stem Cell Technologies) and used to coat coverslips at $37\,^{\circ}C$ for 1 h. Coverslips were then incubated with $10\,mg\,ml^{-1}$ Sodium Borohydride for 1 min, followed by 70% ethanol for 20 min. Three washes with $1\times$ PBS were performed between each step. DMEM media was added to the coverslips at $37\,^{\circ}C$ for 1 h before cell plating.

NMuMG-ErbB2 and HCC1954 cells were pre-treated with TGFβ for 24 h, plated (30,000 cells) onto gelatin-coated coverslips and incubated at $37\,^{\circ}C$ for 24 h. Coverslips were fixed in 4% paraformaldehyde (PFA) and cells were permeabilized with 0.2% Triton X-100, rinsed with 100 mM Glycine in PBS and blocked in 2% BSA in IF buffer (0.2% Triton X-100 and 0.05% Tween-20 in PBS). Antibodies against LPP (1:750; Cat. #: 3389, Cell Signaling), Tks5/FISH (1:500; Cat. #: sc-30122, Santa Cruz) and F-actin/Phalloidin (1:1,000; Cat. #: A22287, ThermoFisher Scientific) were used. Where indicated, DAPI (Cat. #: D3571, Invitrogen) was used to visualize nuclei. Samples were mounted using Immu-Mount (Cat. #: 9990402, Thermo scientific).

Image acquisition and line scan analysis were performed with ZEN imaging software on a Zeiss LSM 510 and LSM710 confocal microscope and a plan-Apochromat $\times63/1.4$ NA oil objective (Carl Zeiss Inc.). Z-stacks were acquired at $0.20\,\mu m$ steps over $5.20\,\mu m$ for HCC1954 model and over $6.0\,\mu m$ for NMuMG-ErbB2 model. Orthogonal views were generated in Imaris v8.3.1 (Bitplane). Quantification of degraded gelatin area was performed using Cell Profiler cell image analysis software (www.cellprofiler.org).

**Gelatin zymography.** MMP2 and MMP9 enzymatic activities were determined by gelatin zymography[64]. CM was collected from TGFβ-treated (24 h) or untreated

cells and spun at 1,200 r.p.m. at 4 °C for 4 min to remove any cells. CM, containing 10 μg of total protein, was mixed with 6 × non-reducing loading dye before loading onto 7.5% polyacrylamide gels containing 0.1% gelatin for separation. Gels were washed with 2.5% Triton X-100 and incubated overnight at 37 °C at 60 r.p.m. in 50 mmol l$^{-1}$ Tris-HCl [pH 7.4]/5 mmol l$^{-1}$ CaCl$_2$/200 mmol l$^{-1}$ NaCl buffer. The reaction was stopped by incubating the gels in 50% methanol & 10% acetic acid for 5 min, followed by staining with Coomassie blue. Gels were visualized using Odyssey Imaging system (Licor). Areas of the gel with active MMPs were observed as white degraded bands and quantified using ImageStudio Lite (Licor). The data represents the average of three independent experiments.

**Mammary fat pad and tail vein injections.** All mammary fat pad and tail vein injections were performed in 5-week-old NCr nude female mice (Taconic Biosciences, Inc.) and 10 mice per cohort were used for each independent experiment. Data quantification was excluded from animals that were euthanized before established end points due to non-tumour burden-related causes, or from animals that died and timely necropsy or sample collection could not be performed.

NMuMG-ErbB2 cells and NIC cells were resuspended in 50:50 mixture of matrigel in PBS (Cat. #: 354234, BD Biosciences) and injected into the number four mammary fat pad of mice. Tumour growth was determined by non-blinded weekly caliper measurements and tumour volumes calculated using the following formula: (width × length$^2$ × π)/6. For Fig. 1, $7.5 \times 10^5$ NMuMG-ErbB2 cells and $3.5 \times 10^5$ NIC cells were used. NMuMG tumours were resected at $450 \pm 50$ mm$^3$ and NIC tumours were resected at $400 \pm 50$ mm$^3$ at 4 weeks post injection. Half the tumour material was flash frozen directly into liquid nitrogen and the other half was fixed in 4% PFA and then embedded in paraffin. Mice were euthanized at the indicated times and all lobes of the lung were harvested. The number of macroscopic metastases on all lobes of the lung was counted. For Supplementary Fig. 4, $2.5 \times 10^6$ NMuMG-ErbB2 cells and $1 \times 10^6$ NIC cells were injected and tumours growth was monitored over 5 weeks where they reached an average volume of 1,500–2,100 mm$^3$. At the time of euthanasia, whole blood was drawn for CTC analysis. Tumour weight was recorded and lung tissues were removed to assess for metastatic burden. For Fig. 10, $2.5 \times 10^6$ NMuMG-ErbB2-derived cell populations were injected into the mammary fat pad and mice were euthanized after 6 weeks. At the time of necropsy, final lung tumour volume and tumour weight were recorded. Whole blood was collected by cardiac puncture, and lung tissue was harvested for metastatic burden analyses.

NMuMG-ErB2 cells ($2 \times 10^5$) were resuspended in PBS and injected into the lateral tail vein of NCr nude mice. Mice were euthanized after the indicated times and the lungs were harvested. The number of macroscopic lesions on all surfaces of the lungs was counted.

Where indicated, mice were given filter-sterilized oral doxycycline (Cat. #: D9891, Sigma-Aldrich) at a concentration of 2 mg ml$^{-1}$, which was diluted in a 5% sucrose solution. Control, vehicle alone, cohorts received a 5% sucrose solution. Dox-administration began 1 week before tumour cell injections and was maintained until the end of the experiment. Dox-containing solutions and vehicle solutions were refreshed twice weekly. NMuMG-ErbB2 breast cancer cells were also pre-treated in vitro with 1 μg ml$^{-1}$ dox (Cat. #: D9891, Sigma-Aldrich) for 1 week before injection.

Mice were housed in facilities managed by the McGill University Animal Resources Centre, and all animal experiments were conducted under a McGill University-approved Animal Use Protocol in accordance with guidelines established by the Canadian Council on Animal Care.

**Histology and immunohistochemistry.** Lungs were collected at necropsy, fixed in 4% paraformaldehyde, and embedded in paraffin. Four-step sections (100 μm per step) were obtained from each set of lungs. The sections were stained with H&E and scanned using a Scanscope XT digital slide scanner (Aperio Technologies). Digital images of lung sections were used to analyse the metastatic burden. Lung tumour lesions were digitally demarcated and the number of lesions per section and the individual lesion area was determined using the Spectrum software (Aperio Technologies). The metastatic burden was calculated as an average of the total area of tumour lesions divided by total lung area across four-step sections.

Immunohistochemical staining for LPP or ErbB2 was performed on paraffin sections. Tissues were subjected to standard heat-induced epitope retrieval, using a steamer, in 10 mM Citrate buffer (pH 6.0). Slides were incubated overnight at 4 °C with a monoclonal rabbit anti-LPP antibody (EPR6478) (1:750 dilution, Cat. #: ab126608; abcam) or a polyclonal rabbit anti-c-ErbB2 antibody (1:750 dilution, Cat. #: A0485; Dako). Following incubation with the primary antibodies, secondary biotin-conjugated antibodies were applied for 30 min. After washing with 1 × PBS, sections were developed with diaminobenzidine (Dako) as the chromogen. All slides were counterstained using Harris haematoxylin.

**Circulating tumour cells.** Tumour-bearing mice were placed under terminal anaesthesia and whole blood was drawn by cardiac puncture into 10 mM EDTA-coated syringes. Blood was transferred into EDTA microtubes (Cat #: 41.1395.105, Sarstedt) and kept on a rocker until ready for processing, within 2 h from withdrawal. Whole blood was diluted at 1:1 ratio with sterile 1X PBS and layered onto

Ficoll-Paque Plus (cat#: 17-1440-02, GE Healthcare) and then spun at 400 g 20 °C for 30 min with no de-acceleration. After gradient separation, the interphase containing mononuclear cells was isolated and washed 3 times with PBS. The cell pellet was then incubated with red blood cell lysis buffer and washed as per manufacturer's protocol (cat#: 555899, BD Pharm Lyse). The final pellet was resuspended in the appropriate cell media and plated onto 5 μg (cm$^2$)$^{-1}$ fibronectin-coated (cat# FC010, Millipore) six-well tissue culture plates.

The next day, cell media was changed and puromycin selection antibiotic at 3 μg ml$^{-1}$ was added. After 14 days, the adherent CTC-derived cell colonies were fixed in formalin for 15 min, and stained with crystal violet for 15 min with thee ddH$_2$0 washes in between each step. Images of the entire well were captured with a × 0.5 objective mounted on a Zeiss microscope (AxioZoom v16) and the number of colonies per image was determined using the Cell Counter plugin in Image J.

**Ex ovo chick chorioallantoic membrane assay.** $5 \times 10^6$ NMuMG-ErbB2 breast cancer cells expressing eGFP-LPP WT, mLIM1 or ΔABD were injected intravenously into 14 days post-fertilization (dpf) avian embryo CAM ($n \geq 3$ per group) using a micropipette syringe[13]. At 4 h post cell injection, 50 μl of lectin-A647 was intravenously injected to label the luminal endothelial surface within the CAM. After 10 min following lectin injection, the proportion of cells in the extravascular space was enumerated in randomly selected fields ($n \geq 10$ per animal). At least 500 breast cancer cells were examined for each cell line. Nikon A1r microscope equipped with × 25/ × 60 objectives and Nikon Elements image acquisition and analysis software were used for all experiments. GraphPad Prism software was used for data plotting and analysis.

**Migration and invasion assays.** NMuMG-derived cell populations, treated with or without TGFβ, were trypsinized and resuspended in serum-free media prior to plating into Boyden chambers (Cat. #: 35–3097, BD Falcon)[17] and 50 μl of 6% Matrigel was used for invasion assays. To quantify the migration and invasion data, five images per well were acquired from each well using a camera-equipped microscope and the × 10 objective. The pixel count for all images was obtained using Scion Image software (Scion Corporation). The data represents the average of 3 to 4 independent experiments performed in duplicate.

**Statistical analysis.** For all in vitro studies, each experiment was performed at least three times. The exact number of independent experiments and replicates are indicated in the figure legends. For in vivo studies, a priori power analysis was performed using the G*Power statistical analysis program[65,66]. By assuming two tails with normal distribution between two groups, calculations revealed that a sample size of $n = 8$ would be sufficient to provide 95% power with 0.05 α-error probability and an effect size of 2. Thus, a sample size of 10 mice per cohort was used in all experiments to account for the potential loss of animals over the duration of the experiment. Statistical significance values (P values) were obtained by performing a two-sample unequal variance student's t-test. Error bars represent s.e.m.

**Data availability.** The data supporting the findings of this study are available within the article or in the Supplementary Information files, and are available upon request.

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

## Acknowledgements

We acknowledge technical assistance from the McGill GCRC Histology core facility for routine histology services and the McGill University Life Sciences Complex Advanced BioImaging Facility (ABIF) for microscopy support. We thank members of the Siegel laboratory for thoughtful discussions and critical reading of the manuscript. We thank Zhifeng Dong for assisting with IHC staining, Yasmina Hachem for helping with lung metastasis burden quantification, and Sébastien Tabariès for lentivirus preparation. We thank Charles Rajadurai and Serhiy Havrylov for helpful advice on setting up the gelatin degradation assays and image quantification. We are grateful for Naila Chughtai's input on isolating circulating tumour cells. E.N. acknowledges salary support from the US Army Department of Defense (Breast Cancer Research Program, grant number W81XWH-11-1-0008) and the RI-MUHC, and P.M.S. holds a William Dawson Scholarship from McGill University. This work was supported by a grant to P.M.S. from the Canadian Cancer Society Research Institute (CCSRI, Grant # 702908).

## Author contributions

E.N. and P.M.S. designed the experiments, interpreted the results and wrote the manuscript. E.N. performed the majority of the experiments included in the manuscript, unless otherwise specified. K.S. performed the CAM extravasation assay, and in conjunction with J.D.L., interpreted the results. H.W.S. and W.J.M. established NIC; Src$^{+/+}$ and NIC; Src$^{fl/fl}$ cell lines used in this study. J.C. assisted in the generation of the stable LPP knockdowns and characterized the NMuMG-ErbB2 and NIC cell populations for their TGFβ responsiveness.

## Additional information

**Competing interests:** The authors declare no competing financial interests.

