## [Peer Review File · Nature Communications]

Reviewers' comments:

Reviewer #1 (Remarks to the Author):

The manuscript of Ngan et al shows that LPP is a Src-family kinase substrate that regulates formation of invadopodia that are required for breast cancer metastasis. This is of significance because understanding the mechanisms regulating invadopodia formation, and function, during cancer metastasis might provide new targets for the development of novel anti-metastatic therapies.

The data are well presented and largely support the conclusions. The manuscript is well written and the experiments are extensive and appear to have been well performed.

The main weakness of this study is that the cell line models used in the in vivo experiments were originally derived experimentally from mice. This leaves the important question of whether the findings are relevant to human breast cancer. It would be important to show that LPP has the same role in ERB2+ human breast cancer cells.

Additional points

1. LPP promotes breast cancer cell intravasation p 6-7. In this section the authors use colony formation assays to measure changes in CTCs from knock-down vs control mice. They state that lower colony formation shows less circulating tumour cells (CTC) but is this necessarily the case? This could reflect a change cell phenotype. Colony forming ability is often used as a measure of stemness for example so it could be that the cells are simply less able to form colonies. An actual count of CTCs would be a more accurate assessment of number of cells in the blood stream. In any event the reduction in CTC formation is not necessarily due to a loss of intravasation.

2. p 12. Similar to above - how do the authors know that the tumour cells are failing to arrive at the lung and are not incapable of forming detectable metastasis? Are micro-metastases present?

2. It would be logical to include data in Fig 7 with Fig. 1 as it is essentially the control experiments to show the metastasis is indeed LPP-specific, and not due to off-target effects.

3. The authors conclude that LPP promotes metastasis of ErbB2 expressing cells but should perhaps change this to specifically say lung metastasis, as no other metastasis was assessed

Reviewer #2 (Remarks to the Author):

In this manuscript, Ngan et al evaluate LPP, a protein that has already been shown to be involved in breast cancer migration and invasion. They conclude that LPP is required for invadopodia formation in vitro and extravasation in vivo using the CAM assay. They also link LPP to Src signaling. There are several technical issues with the manuscript, listed below. In addition, several of the conclusions are overstated. Finally, there is concern that even if these issues are resolved, there is insufficient new information on either LPP or invadopodia to justify publication in this journal.

1. Throughout the manuscript, the authors refer to invadopodia function, when the assay they are using measures matrix degradation. Of course, invadopodia are associated with matrix degradation, but it is not at all clear that this is the only mechanism by which this happens. It is also possible to dissociate invadopodia formation and function. Assays for both should be conducted.

2. Many experiments are conducted with a single shRNA experiment with no rescue, raising the

concern about off-target effects. This needs to be remedied.

3. The quantification of gelatin degradation (as regions) is not the accepted measure in the field.

4. If Tks5 is "a specific and early marker of invadopodia", then why are structures lacking Tks5, but containing LPP, still referred to as invadopodia (Figure 5b)? Furthermore, the quantification of more images than the ones shown (with just 2-3 representative cells) would be needed to support any conclusion about the effect of LPP mutants on protrusions in vivo.

5. In Figure 6, the LPP appears to be expressed throughout the cell, and it is not appropriate to call any protrusions invadopodia, absent a specific marker.

6. No data are provided for the kinetics of LPP knockdown upon doxycycline treatment (Figure 7). Typically, it takes days to see a change in protein expression after dox treatment, by which time the cells will have already extravasated. In this case, the assay essentially repeats what was already shown in Figure 1.

7. Both dasatinib and PP2 are far from selective for Src family kinases, and confirmation of these data with, for example, knockdown experiments, would be needed.

8. In vivo analysis of the 5F mutants should be performed to complement the in vitro data.

9. In the discussion, the authors refer to LPP recruiting Tks5. Where are the data to support that statement?

10. Proposing a model in which LPP is involved in force generation in invadopodia is also premature in the absence of any supporting data.

Reviewer #1:

We are please that the reviewer appreciated the significance of our findings and felt that “*The data are well presented and largely support the conclusions.*” and “*The manuscript was well written and the experiments are extensive and appear to have been well performed*” The reviewer has made several suggestions to improve the manuscript, which we discuss below.

1) The main weakness of this study is that the cell line models used in the *in vivo* experiments were originally derived experimentally from mice. This leaves the important question of whether the findings are relevant to human breast cancer. It would be important to show that LPP has the same role in ERB2+ human breast cancer cells.

We thank the reviewer for this comment. Indeed, our previous submission focused on two independent murine ErbB2-expressing breast cancer cell models. We have now included a human ErbB2+ breast cancer model (HCC1954) in which we show that knockdown of LPP results in diminished gelatin degradation (Figure 3b of the revised manuscript) and impaired invadopodia formation (Figure 4 of the revised manuscript). We have previously shown that LPP is required for TGFβ-induced migration and invasion in HCC1954 cells (Ngan et al., J Cell Sci. 2013, 126(Pt 9):1981-91). Thus, these data clearly show that LPP functions in a similar manner in both mouse and human ErbB2-expressing breast cancer cells to control invadopodia formation and cell invasion.

Additional points:

1) LPP promotes breast cancer cell intravasation (p6-7). In this section the authors use colony formation assays to measure changes in CTCs from knockdown vs control mice. They state that

lower colony formation shows less circulating tumour cells (CTC) but is this necessarily the case? This could reflect a change cell phenotype. Colony forming ability is often used as a measure of stemness for example so it could be that the cells are simply less able to form colonies. An actual count of CTCs would be a more accurate assessment of number of cells in the blood stream.

We apologize to the reviewer if our description of this experiment was not clear. The data presented in Figure 2 of the revised manuscript are colonies that grew following isolation of circulating tumor cells from the peripheral blood of tumor bearing mice (cardiac punctures). Thus, the experiment proposed by the reviewer was indeed the experiment that we included in the original submission. We demonstrate that a reduction of LPP expression in both NMUMG-ErbB2 and NIC breast cancer cells leads to reduced numbers of circulating tumor cells in tumor bearing mice.

2) p 12. Similar to above – how do the authors know that the tumour cells are failing to arrive at the lung and are not incapable of forming detectable metastasis? Are micro-metastases present?

In the case of the spontaneous metastasis model (mammary fat pad injections) (Figure 1, Figure 2 and Supplementary Fig. S4 of the revised manuscript), we show that reduced LPP expression has no impact on primary tumor growth. However, the numbers of circulating tumor cells in the bloodstream of tumor-bearing mice is greatly reduced when LPP expression is diminished. These data clearly support the conclusion that reduced LPP expression diminishes the ability of breast cancer cells to intravasate into circulation from the primary tumor. In the context of the experimental metastasis experiments (tail vein injections; Figure 6 of the revised manuscript), we are seeding the cancer cells directly in the lungs, as this is the first capillary bed the cancer cells encounter after injection. Given that loss of LPP does not impair tumor cell proliferation or viability, we interpret the diminished lung metastasis burden following a tail vein injection to mean that the cancer cells were impaired for their ability to extravasate from circulation into the lung parachyma. In our examination of the H&E stained lung sections, we observed a reduction in the number of metastases when LPP levels were reduced; however, we did not detect an increase in the presence of micro-metastases.

3) It would be logical to include data in Fig 7 with Fig. 1 as it is essentially the control experiments to show the metastasis is indeed LPP-specific, and not due to off-target effects.

We disagree with the reviewer that the spontaneous metastasis data (Figure 1 of the revised manuscript) and the experimental metastasis data (Figure 6 of the revised manuscript) should be merged, as they are asking slightly different questions. The data in Figure 1 shows that diminished LPP levels reduces metastases, which could be due to defects in both intravasation and extravasation. The data in Figure 6 indicates that reduced LPP levels impair metastasis by diminishing the ability of breast cancer cells to extravasate. Thus, we would prefer to leave Figure 6 as a stand-alone figure.

4) The authors conclude that LPP promotes metastasis of ErbB2 expressing cells but should perhaps change this to specifically say lung metastasis, as no other metastasis was assessed.

We would argue that loss of LPP would have similar effects on the ability of breast cancer cells to spread to multiple locations other than the lungs. Indeed, we demonstrate that LPP function is critical for invadopodia formation, which is a basic requirement of metastatic cells regardless of the site in which they establish. This is reflected by the fact that circulating tumor cell numbers are diminished in two independent ErbB2-expressing models. In the current manuscript, the tumor models happen to preferentially spread to the lungs; however, this does not preclude a role for LPP in cancer cell dissemination to other sites. However, in light of the reviewer's comments we have changed the title as follows "LPP is a Src substrate required for invadopodia formation and efficient breast cancer lung metastasis".

Reviewer #2:

We thank the reviewer for constructive criticisms of our work and provide a detailed response to the points raised during the initial review.

1) The reviewer raises the concern that "*there is insufficient new information on either LPP or invadopodia to justify publication in this journal*".

We respectfully disagree with the reviewer's assertion that insufficient new information is provided in the manuscript to warrant publication. There are numerous aspects of the current work that are completely novel and that have not been described in the published literature.

1) *While we were the first to implicate LPP in modulating the migratory and invasive characteristics of breast cancer cells (Ngan et al., J Cell Sci. 2013, 126(Pt 9):1981-91), the current manuscript extends these data to show that LPP is a critical regulator of breast cancer metastasis to the lung, while being dispensable for primary tumor growth. This is an important distinction as in vitro assays are only surrogates for the metastatic process in vivo. **Thus, we are the first to show that LPP is an important promoter of breast cancer metastasis in vivo.***

2) *LPP has been previously implicated as a focal adhesion protein that controls cell migration in diverse cell types. **However, our manuscript is the very first to demonstrate that LPP also co-localizes to invadopodia and is functionally involved in matrix degradation.***

3) ***We show are the first to demonstrate that LPP is a substrate for Src and that LPP phosphorylation at specific sites are necessary for LPP-mediated invadopodia formation, matrix degradation and metastasis.** We also demonstrate that LPP phosphorylation is specifically required for cancer cell invasion and is dispensable for migration, which represents a unique observation.*

2) Throughout the manuscript, the authors refer to invadopodia function, when the assay they are using measures matrix degradation. Of course, invadopodia are associated with matrix degradation, but it is not at all clear that this is the only mechanism by which this happens. It is

also possible to dissociate invadopodia formation and function. Assays for both should be conducted.

We have added considerable data to the manuscript to ascertain whether LPP is localized to invadopodia, using well-established and accepted markers (Tks5, actin co-localization). We now conclusively demonstrate that LPP can be found co-localized in defined structures that are positive for actin and Tks5, which coincide with areas of matrix degradation using two independent ErbB2 positive cells models (one human [HCC1954] and one mouse [NMuMG-ErbB2]) (see Figure 4 and Figure S6 in the revised manuscript). These data clearly show that LPP is present within invadopodia and that LPP function is required for matrix degradation.

3) Many experiments are conducted with a single shRNA experiment with no rescue, raising the concern about off-target effects. This needs to be remedied.

While the reviewer is correct in the statement that one shRNA targeting LPP was used in the mouse systems, we have shown in each instance that the observed phenotypes resulting from the reduction of LPP expression is rescued by expression of wild type LPP. This is the gold standard for demonstrating that the effects we describe are not the result of "off-target" effects. We show that diminished gelatin degradation observed with LPP knockdown is rescued with wild type LPP (see Figure 3a and Figure 8 in the revised manuscript). We show that diminished lung metastasis observed in cells with reduced LPP can be fully rescued by expression of wild type LPP (see Figure 6 and Figure 10 in the revised manuscript). We show that the reduction in TGF β -induced migration and invasion seen in LPP knockdown cells can be rescued by wild type LPP (see Figure 8 and Figure 9 in the revised manuscript). Finally, we demonstrate that the reduction in circulating tumor cells that is observed when LPP expression is diminished can be fully rescued by expression of wild type LPP (see Figure 10c in the revised manuscript). Thus, we have demonstrated that each metastatic phenotype that is impaired by loss of LPP can be rescued by expression of wild type LPP and confirms that we are not observing off-target effects of the LPP shRNA.

4) The quantification of gelatin degradation (as regions) is not the accepted measure in the field.

Our initial submission contained both the number of degraded areas as well as the total area degraded as independent measures, the latter is the accepted measure of invadopodia-mediated matrix degradation. In response to the reviewer's comment, we have removed the number of degraded areas in the revised manuscript (Figure 3 and Figure 8).

5) If Tks5 is "a specific and early marker of invadopodia", then why are structures lacking Tks5, but containing LPP, still referred to as invadopodia (Figure 5b)? Furthermore, the quantification of more images than the ones shown (with just 2-3 representative cells) would be needed to support any conclusion about the effect of LPP mutants on protrusions *in vivo*.

These data were quite interesting and would suggest that LPP localization to invadopodia may be an earlier event than Tks5, and that LPP may be required to recruit Tks5 to invadopodia. However, we were unable to perform live imaging on cells expressing eGFP-LPP and Tks5 to show that LPP was indeed recruited prior to Tks5 localization at nascent invadopodia. We have modified the text to diminish the speculation on this point in the discussion. The main point of the CAM assay was to show, using an independent system where invadopodia are critical for extravasation, that LPP is required for efficient extravasation. This is clearly shown by the fact that LPP-expressing cells extravasate easily while cells that express the LPP mutants are severely impaired. This data represent the analysis of hundreds of cells for each construct and convincingly demonstrate a role for LPP in mediating cancer cell extravasation in an independent in vivo system. The images showing individual cells as they extravasate are not key pieces of data and have been moved to Supplemental Figure S8 of the revised manuscript.

6) In Figure 6, the LPP appears to be expressed throughout the cell, and it is not appropriate to call any protrusions invadopodia, absent a specific marker.

We agree with the reviewer that additional data was required from our original submission to demonstrate that LPP is a novel constituent of invadopodia. We have invested considerable effort to show that LPP is indeed co-localized with accepted markers of invadopodia (Tks5 and actin) at sites of gelatin degradation in two independent ErbB2-expressing cancer cell models (HCC1954 [human] and NMuMG-ErbB2 [mouse] cells) (see Figure 4 and Figure S6 in the revised manuscript). This augments the co-localization of LPP and Tks5 that we observed in the CAM assay (see Figure 5 of the revised manuscript).

7) No data are provided for the kinetics of LPP knockdown upon doxycycline treatment (Figure 7). Typically, it takes days to see a change in protein expression after dox treatment, by which time the cells will have already extravasated. In this case, the assay essentially repeats what was already shown in Figure 1.

We apologize to the reviewer that the setup for this experiment was unclear; however, it was indicated in the methods section of the original submission that the NMuMG-ErbB2 breast cancer cells were cultured in doxycycline for 1 week prior to injection to ensure that endogenous levels of LPP were diminished. To convince the reviewer that this is indeed the case, we have included an immunoblot showing endogenous levels of LPP following 4 days of doxycycline treatment of cells prior to injection (following an additional 3 days in the presence of doxycycline). This data clearly shows that the knockdown has worked well and that endogenous LPP levels are significantly reduced prior to injection into mice (see Figure 6, panel a in the revised manuscript). To ensure that LPP levels remain low, the cells were injected into mice that had been administered doxycycline prior to tumor cell injection. This assay (Figure 6) is significantly different than the data presented in Figure 1, which shows spontaneous metastasis

from the primary site. The data shown in Figure 6 of the revised manuscript is an experimental metastasis assay in which the breast cancer cells are injected into the lateral tail vein. The data in Figure 1 shows that reduced LPP results in diminished lung metastasis formation, which could result from fewer cells reaching the circulation (intravasation defect) or fewer cells exiting the bloodstream into the lung (extravasation). The data shown in Figure 6 reveals that LPP knockdown cells have a clear defect in extravasation. Using a combination of these two approaches, we have been able to determine that LPP functions to promote both intravasation and extravasation

8) Both dasatinib and PP2 are far from selective for Src family kinases, and confirmation of these data with, for example, knockdown experiments, would be needed.

We agree with the reviewer that these two Src family kinase inhibitors can target other kinases. To address the reviewer's concerns, we took three approaches. First, we generated Src-specific knockdowns using two independent shRNAs in the NMuMG-ErbB2 breast cancer cells and show that diminished Src expression leads to a block in TGF β -induced LPP phosphorylation (see Figure 7b in the revised manuscript). Second, we overexpressed a constitutively active form of Src (SrcY529F) in NMuMG-ErbB2 cells and demonstrated that baseline phosphorylation of LPP was increased in the presence of active Src, which was further elevated in response to TGF β (see Figure 7c in the revised manuscript). These two pieces of data reveal that Src is both necessary and sufficient to induce LPP tyrosine phosphorylation in NMuMG-ErbB2 cells. For the third approach, we took advantage of NIC breast cancer cells containing floxed Src alleles, which allowed us to compare NIC breast cancer cells that retained Src expression versus those in which Src was specifically deleted via cre-mediated excision. These data reveal that LPP is phosphorylated following TGF β stimulation in cells expressing Src, a response that was eliminated in NIC cells deficient in Src (see Figure 7d in the revised manuscript). Together, these data convincingly demonstrate that Src can mediate the tyrosine phosphorylation of LPP in response to TGF β treatment.

9) *In vivo* analysis of the 5F mutants should be performed to complement the *in vitro* data.

*We agree with the reviewer that assessment of specific LPP phosphorylation sites on the ability of breast cancer cells to metastasize is an important experiment. We have actually performed additional experiments that have mapped three specific tyrosine residues within LPP that are important for TGF β -induced invasion. Interestingly, a mutant of LPP in which three tyrosine residues are mutated to phenylalanine residues (Y245F/Y301F/Y302F) completely ablated the ability of TGF β to induce breast cancer invasion (see Figure 9 of the revised manuscript). Interestingly, these same mutations do not impair TGF β -mediated migration (see Figure 9 of the revised manuscript), indicating that these mutants remain functional with respect to certain LPP functions. We have further extended these analyses *in vivo*, where we demonstrate that LPP-WT, two phosphorylation mutants that behave like LPP-WT (LPP-Y245F,*

LPP-Y245F/Y297F/Y298F) and the LPP phosphorylation mutant exhibiting defective invasion in vitro (LPP-Y245F/Y301F/Y302F) were introduced into NMuMG-ErbB2 cells with reduced endogenous LPP. Importantly, we show that primary tumor growth was unaffected by the loss of LPP or by expression of any of the mutant LPP constructs. Interestingly, only the LPP-Y245F/Y301F/Y302F mutant exhibited diminished circulating tumor cells in circulation and reduced formation of lung metastases in vivo. Taken together, we have further narrowed down the requirement for tyrosine phosphorylation of residues Y245, Y301 and Y302 within LPP for efficient breast cancer cell invasion and lung metastasis.

10) In the discussion, the authors refer to LPP recruiting Tks5. Where are the data to support that statement?

We agree that speculation that LPP may function to recruit Tks5 is premature and would require live-cell imaging of invadopodia as they form to substantiate this statement. We have removed this sentence from the discussion in the revised manuscript.

11) Proposing a model in which LPP is involved in force generation in invadopodia is also premature in the absence of any supporting data.

We agree with the reviewer that such a mechanism is purely speculative at this stage and have removed this section from the discussion of the revised manuscript.

REVIEWERS' COMMENTS:

Reviewer #1 (Remarks to the Author):

The authors have addressed the main concern regarding the relevance of the findings in human breast cancer. Substantial additional experiments have been included in the revised manuscript, which is now much improved.

Reviewer #2 (Remarks to the Author):

I think the authors have addressed the technical comments raised. But I still have concerns about the novelty of the paper. What is provided is a description of another protein that can localize to invadopodia and is a Src substrate. Since invadopodia are known to be involved in intra- and extravasation of cancer cells, it is perhaps not unexpected that LPP is also required for this. What is lacking are novel mechanistic insights into how LPP accomplishes this. Discussion points in the previous version that it might sense mechanical force or recruit Tks5 might have provided that novelty, if data had been provided. As it stands however, we do not gain much new insight into how invadopodia control invasive behavior.

Reviewer #1:

The authors have addressed my main concern regarding the relevance of the findings in human breast cancer. Substantial additional experiments have been included in the revised manuscript, which is now much improved.

Reviewer #2:

I think the authors have addressed the technical comments raised. But I still have concerns about the novelty of the paper. What is provided is a description of another protein that can localize to invadopodia and is a Src substrate. Since invadopodia are known to be involved in intravasation and extravasation of cancer cells, it is perhaps not unexpected that LPP is also required for this. What is lacking are novel mechanistic insights into how LPP accomplishes this. Discussion points in the previous version that it might sense mechanical force or recruit Tks5 might have provided that novelty, if data had been provided. As it stands however, we do not gain much new insight into how invadopodia control invasive behavior.

We respectfully disagree with the reviewer on this point. The novel aspects of current manuscript include 1) the first demonstration that LPP is an important promoter of breast cancer metastasis in vivo (previous work has been restricted to its role in cell migration and invasion in vitro), 2) the first demonstration that LPP is a constituent of invadopodia (previous studies have only examined LPP in the context of focal adhesions) and 3) the identification of specific Src-dependent phosphorylation sites on LPP that are critical for LPP-mediated invadopodia formation, matrix degradation and metastasis. While we agree with the reviewer that deciphering the precise molecular mechanisms through which LPP functions to form invadopodia, we feel that such avenues of investigation is beyond the scope of the current publication and will constitute the focus of subsequent publications. Indeed, the potential role of LPP as a mechanosensor during cell migration and invasion (invadopodia formation) is an avenue that is currently under investigation in the laboratory. In addition, we are interested in identifying LPP interaction networks that are lost with LPP phosphorylation mutants that can no longer support breast cancer invasion and metastasis. We acknowledge the need for further experimentation to provide the detailed mechanistic insights to these interesting questions in the discussion section of the manuscript (Page 22, line 448-450; Page 23, line 470-472).